# Astrogliosis and neuroinflammation underlie scoliosis upon cilia dysfunction

Morgane Djebar[1], Isabelle Anselme[1], Guillaume Pezeron[2], Pierre-Luc Bardet[1], Yasmine Cantaut-Belarif[3], Alexis Eschstruth[1], Diego López-Santos[1], Hélène Le Ribeuz[1], Arnim Jenett[4], Hanane Khoury[1], Joelle Veziers[5], Caroline Parmentier[6], Aurélie Hirschler[7], Christine Carapito[7], Ruxandra Bachmann-Gagescu[8,9], Sylvie Schneider-Maunoury[1]*, Christine Vesque[1]*

[1]Sorbonne Université, CNRS UMR7622, INSERM U1156, Institut de Biologie Paris Seine (IBPS) - Developmental Biology Unit, Paris, France; [2]Molecular Physiology and Adaptation (PhyMA - UMR7221), Muséum National d'Histoire Naturelle, CNRS, Paris, France; [3]Institut du Cerveau et de la Moelle épinière (ICM), Sorbonne Université, Inserm U 16 1127, CNRS UMR 7225, F-75013, Paris, France; [4]TEFOR Paris-Saclay, CNRS UMS2010 / INRA UMS1451, Université Paris-Saclay, Paris, France; [5]Inserm UMR 1229, CHU Nantes PHU4 OTONN, SC3M facility, Inserm UMS 016, CNRS 3556, Université de Nantes, Nantes, France; [6]Sorbonne Université, CNRS UMR8246, INSERM U1130, Institut de Biologie Paris Seine (IBPS) – Neurosciences Paris Seine (NPS), Paris, France; [7]Laboratoire de Spectrométrie de Masse Bio-Organique, IPHC, UMR 7178, 23 Université de Strasbourg, CNRS, Infrastructure Nationale de Protéomique ProFI - 24 FR2048, Strasbourg, France; [8]Institute of Medical Genetics, University of Zurich, Zurich, Switzerland; [9]Institute of Molecular Life Sciences, University of Zurich, Zurich, Switzerland

**\*For correspondence:**
sylvie.schneider-maunoury@
sorbonne-universite.fr (SS-M);
christine.vesque@upmc.fr (CV)

**Competing interest:** The authors declare that no competing interests exist.

**Abstract** Cilia defects lead to scoliosis in zebrafish, but the underlying pathogenic mechanisms are poorly understood and may diverge depending on the mutated gene. Here, we dissected the mechanisms of scoliosis onset in a zebrafish mutant for the *rpgrip1l* gene encoding a ciliary transition zone protein. *rpgrip1l* mutant fish developed scoliosis with near-total penetrance but asynchronous onset in juveniles. Taking advantage of this asynchrony, we found that curvature onset was preceded by ventricle dilations and was concomitant to the perturbation of Reissner fiber polymerization and to the loss of multiciliated tufts around the subcommissural organ. Rescue experiments showed that Rpgrip1l was exclusively required in *foxj1a*-expressing cells to prevent axis curvature. Genetic interactions investigations ruled out Urp1/2 levels as a main driver of scoliosis in *rpgrip1* mutants. Transcriptomic and proteomic studies identified neuroinflammation associated with increased Annexin levels as a potential mechanism of scoliosis development in *rpgrip1l* juveniles. Investigating the cell types associated with *annexin2* over-expression, we uncovered astrogliosis, arising in glial cells surrounding the diencephalic and rhombencephalic ventricles just before scoliosis onset and increasing with time in severity. Anti-inflammatory drug treatment reduced scoliosis penetrance and severity and this correlated with reduced astrogliosis and macrophage/microglia enrichment around the diencephalic ventricle. Mutation of the *cep290* gene encoding another transition zone protein also associated astrogliosis with scoliosis. Thus, we propose astrogliosis induced by perturbed ventricular homeostasis and associated with immune cell activation as a novel pathogenic mechanism of zebrafish scoliosis caused by cilia dysfunction.

## eLife assessment

This **valuable** study analyzes the role of rpgrip1l encoding a ciliary transition zone component in the development of neuroinflammation and scoliotic phenotypes in zebrafish. Through proteomic and experimental validation in vivo, the authors demonstrated increased Annexin A2 expression and astrogliosis in the brains of scoliosis fish. Anti-inflammatory drug treatment restored normal spine development in these mutant fish, thus providing additional **convincing** evidence for the role of neuroinflammation in the development of scoliosis in zebrafish.

## Introduction

Idiopathic scoliosis (IS) is a 3D rotation of the spine without vertebral anomalies that affects about 3% of adolescents worldwide. Its etiology remained mysterious, mainly because of the lack of appropriate animal models and the complexity of its inheritance profile in large families presenting a severe scoliosis trait (*Patten et al., 2015*). Since the description of the first IS model in the zebrafish *ptk7* gene mutant (*Hayes et al., 2014*), a number of mutants for genes encoding ciliary proteins were shown to develop scoliosis at late larval and juvenile stage (4–12 weeks post-fertilization, wpf) without any vertebral fusion or fracture, highlighting the link between cilia function and straight axis maintenance in that species (*Grimes et al., 2016*; *Wang et al., 2022*).

Cilia are microtubular organelles with sensory and/or motile functions. Zebrafish mutants in genes involved in cilia motility or in intraflagellar transport display severe embryonic axis curvature and are usually not viable beyond larval stages (*Grimes et al., 2016*; *Rose et al., 2020*). In contrast, mutants in genes implicated in ciliary gating or ciliary trafficking encoding respectively components of the transition zone (TZ) or of the BBsome complex survive to adult stage and develop scoliosis with variable penetrance (*Bentley-Ford et al., 2022*; *Masek et al., 2022*; *Song et al., 2020*; *Wang et al., 2022*). The molecular basis for these differences is still poorly understood, even if progress has been made in our understanding of zebrafish scoliosis. Cilia-driven movements of the cerebrospinal fluid (CSF) are involved in zebrafish axis straightness, both in embryos and juveniles (*Boswell and Ciruna, 2017*; *Grimes et al., 2016*) and are tightly linked to the assembly and maintenance of the Reissner fiber (RF), a SCO-spondin polymer secreted by the subcommissural organ (SCO) and running down the brain and spinal cord CSF-filled cavities (*Cantaut-Belarif et al., 2018*). RF loss at embryonic stage in null *sspo* mutants generates ventral curvatures of the posterior axis of the body and is lethal while its loss at juvenile stage in hypomorphic *sspo* mutants triggers scoliosis with full penetrance (*Cantaut-Belarif et al., 2018*; *Rose et al., 2020*; *Troutwine et al., 2020*). Signaling downstream of the RF in embryos implicates Urp1 and Urp2, two neuropeptides of the Urotensin 2 family expressed in ventral CSF-contacting neurons (CSF-cNs), which act on dorsal muscles in embryos and larvae (*Cantaut-Belarif et al., 2020*; *Gaillard et al., 2023*; *Lu et al., 2020*; *Zhang et al., 2018*). Their combined mutations or the mutation of their receptor gene *uts2r3* lead to scoliosis at adult stages (*Bearce et al., 2022*; *Gaillard et al., 2023*; *Zhang et al., 2018*). However, whether RF maintenance and Urp1/2 signaling are perturbed in juvenile scoliotic mutants of genes encoding TZ proteins ('TZ mutants') and how these perturbations are linked to scoliosis is unknown, especially because TZ mutants do not display any sign of embryonic or larval cilia motility defects (*Wang et al., 2022*). Moreover, two scoliotic models develop axial curvature while maintaining a RF showing that their curvature onset occurs independently of RF loss (*Meyer-Miner et al., 2022*; *Xie et al., 2023*). Finally, neuro-inflammation has been described in a small subset of zebrafish IS models for which anti-inflammatory/anti-oxidant treatments (with NAC or NACET) partially rescue scoliosis penetrance and severity (*Rose et al., 2020*; *Van Gennip et al., 2018*).

In this paper, we dissect the mechanisms of scoliosis appearance in a novel deletion allele of the zebrafish TZ *rpgrip1l* gene. *RPGRIP1L* mutations have been found in neurodevelopmental ciliopathies, Meckel syndrome (MKS) and Joubert syndrome (JS; *Delous et al., 2007*; *Arts et al., 2007*) characterized by severe brain malformations. Other organs such as the kidney and liver are variably affected. Some JS patients also present with severe scoliosis, but this phenotypic trait is of variable penetrance (*Delous et al., 2007*; *Brancati et al., 2008*). Functional studies performed in mouse and zebrafish have shown that *Rpgrip1l* is required for proper central nervous system patterning due to complex perturbations of Sonic Hedgehog signaling (in mice) and cilia planar positioning (in mice and zebrafish; *Andreu-Cervera et al., 2021*; *Mahuzier et al., 2012*; *Vierkotten et al., 2007*). Zebrafish

*rpgrip1l* mutants are viable and develop an axis curvature phenotype at juvenile stages with nearly full penetrance. Using a variety of approaches, we show that (i) Rpgrip1l is required in ciliated cells surrounding ventricular cavities for maintaining a straight axis; (ii) at juvenile stages, *rpgrip1l* mutants develop cilia defects associated with ventricular dilations and loss of the RF; (iii) increased *urp1/2* expression in *rpgrip1l* mutants does not account for spine curvature defects; and (iv) a pervasive astrogliosis process associated with neuroinflammation participates in scoliosis onset and progression. Finally, we show that astrogliosis is conserved in another zebrafish ciliary TZ mutant, *cep290*, identifying a novel mechanism involved in scoliosis upon ciliary dysfunction.

## Results

### *Rpgrip1l*[-/-] fish develop scoliosis asynchronously at juvenile stage

To study the mechanisms of scoliosis appearance upon ciliary dysfunction in zebrafish, we made use of a viable zebrafish deletion mutant in the *rpgrip1l* gene encoding a ciliary transition zone protein (*rpgrip1l*[Δ], identified in ZFIN as *rpgrip1l*[bps1], *Figure 1—figure supplement 1A*). We produced a 26 kb deletion after a CRISPR-Cas9 double-cut which removes 22 out of the 26 exons of the protein (*Figure 1—figure supplement 1A*) and triggers protein truncation at amino-acid (aa) position 88 out of 1256. 90–100% of *rpgrip1l*[Δ/Δ] embryos and larvae were straight (0–10%, depending on the clutch, displayed a sigmoid curvature). They did not display any additional defects found in many ciliary mutant embryos or larvae such as randomized left-right asymmetry, kidney cysts, or retinal anomalies (*Bachmann-Gagescu et al., 2011*; *Kramer-Zucker et al., 2005*; *Tsujikawa and Malicki, 2004*; *Figure 1A* and *Figure 1—figure supplement 1B, D, E*). Another allele with a frameshift mutation in exon 4 (*rpgrip1l*[ex4], identified in ZFIN as *rpgrip1l*[bps2]) showed a similar phenotype (*Figure 1—figure supplement 1A, B*). Considering the large size of the deletion, we considered the *rpgrip1l*[Δ] allele as a null and called it *rpgrip1l*[-/-] thereafter. *rpgrip1l*[-/-] animals developed scoliosis during juvenile stages. Scoliosis appearance was asynchronous between clutches, from 4 weeks post fertilization (wpf) (around 1 cm length) to 11 wpf (1,8–2,3 cm), and also within a clutch. It started with slight upward bending of the tail (the *tail-up* phenotype) and progressed toward severe curvature (*Figure 1A and B*) with 90% penetrance in adults (100% in females and 80% in males; *Figure 1C*). Micro-computed tomography (μCT) at two different stages (5 wpf and 23 wpf) confirmed that spine curvature was three-dimensional and showed no evidence of vertebral fusion, malformation, or fracture (*Figure 1D–G* and *Figure 1—video 1*).

Thus, the ciliary *rpgrip1l*[-/-] mutant shows almost totally penetrant, late-onset juvenile scoliosis without vertebral anomalies, making it a valuable model for studying the etiology of human idiopathic scoliosis. Furthermore, the asynchronous onset of curvature is an asset for studying the chronology of early defects leading to scoliosis.

### Reintroducing RPGRIP1L in foxj1a lineages suppresses scoliosis in *rpgrip1l*[-/-] fish

Since *rpgrip1l* is ubiquitously expressed in zebrafish (*Mahuzier et al., 2012*), scoliosis could have different tissue origins, including a neurological origin as shown for the *ptk7* and *ktnb1* scoliotic fish (*Meyer-Miner et al., 2022*; *Van Gennip et al., 2018*). Indeed, no tissue-specific rescue has been performed in mutants for zebrafish ciliary TZ or BBS genes. We tested if introducing by transgenesis a tagged human RPGRIP1L protein under the control of the col2a1a (*Dale and Topczewski, 2011*) or foxj1a (*Van Gennip et al., 2018*) enhancers (*Figure 1H*) would reduce scoliosis penetrance and severity. The *foxj1a* enhancer drives expression in motile ciliated cells, among which ependymal cells lining CNS cavities (*Grimes et al., 2016*), while the *col2a1a* regulatory region drives expression in cartilage cells including intervertebral discs and semi-circular canals, tissues that have been proposed to be defective in mammalian scoliosis models (*Hitier et al., 2015*; *Karner et al., 2015*). After selecting F1 transgenic fish that expressed the Myc-tagged RPGRIP1L protein in the targeted tissue at 2.5 dpf, we introduced one copy of each transgene into the *rpgrip1l*[-/-] background. The Foxj1a:5XMycRPGRIP1L transgene was sufficient to fully suppress scoliosis in *rpgrip1l*[-/-] fish: none of the transgenic *rpgrip1l*[-/-] fish became scoliotic while virtually all the non-transgenic *rpgrip1l*[-/-] siblings did (*Figure 1I and J*). In contrast, RPGRIP1L expression triggered by the *col2a1a* promoter did not

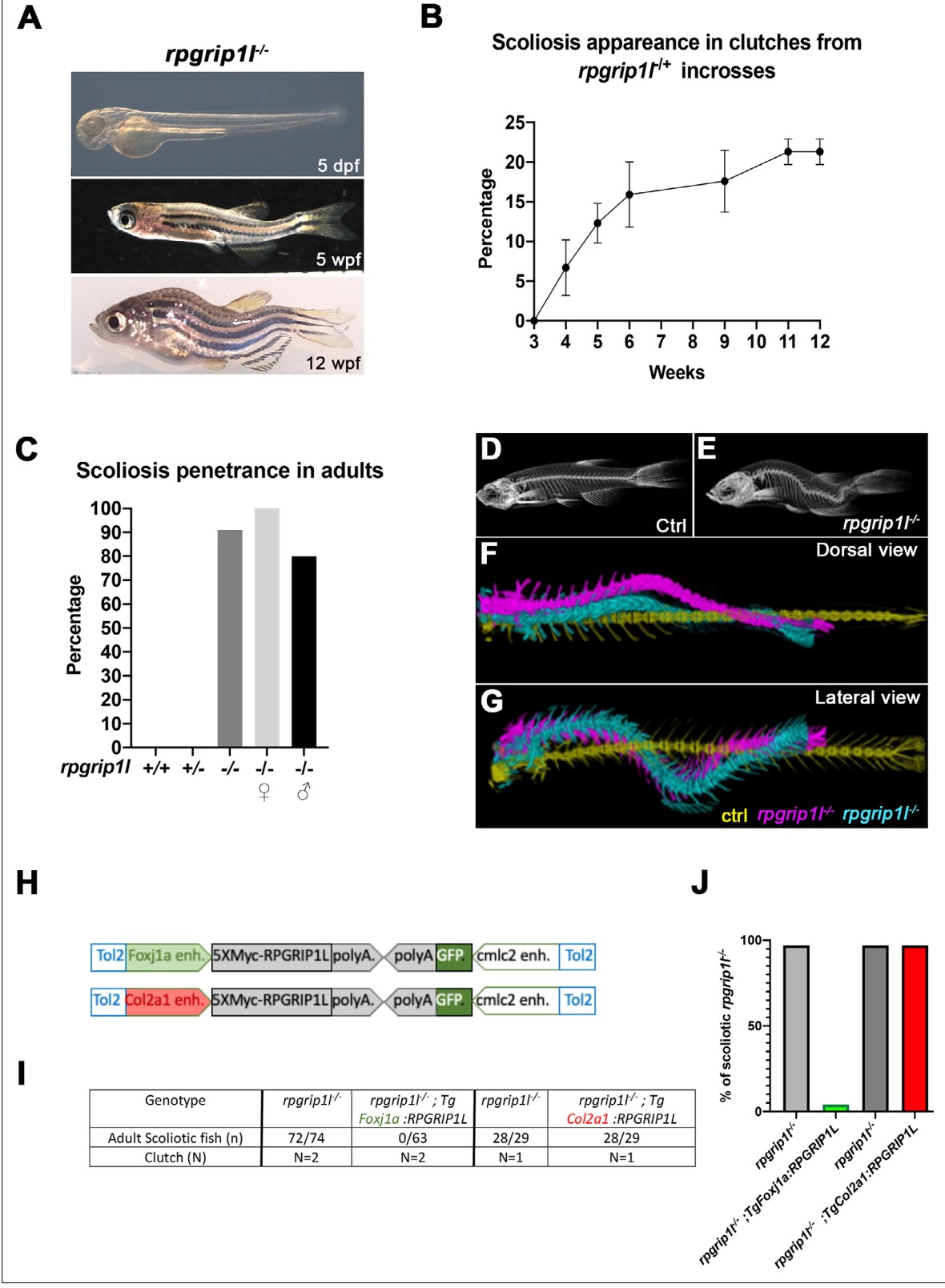

**Figure 1.** *rpgrip1l⁻/⁻* juvenile fish develop scoliosis, which is rescued by RPGRIP1L expression in *foxj1a*-positive cells. (**A**) Representative *rpgrip1l⁻/⁻* fish at 5 days post-fertilization (dpf) (4 mm body length, larvae), 5 weeks pf (wpf) (1.2 cm body length, juveniles) and 12 wpf (2.5 cm body length, adults), showing absence of morphological defects in embryos, onset of spine curvature (tail-up) in juveniles and scoliosis in adults. (**B, C**) Graph showing the time course of scoliosis appearance (**B**) in *rpgrip1l⁻/⁺* incrosses (total 4 clutches, 252 fish) and scoliosis penetrance (**C**) in adults. (**D, E**) Micro-computed

*Figure 1 continued*

tomography (µCT) scans of control siblings (**D**) and *rpgrip1l*$^{-/-}$ (**E**) adult fish. Four fish were analyzed for each condition. (**F, G**) Dorsal and lateral superimposed µCT views of the spines of one control (yellow) and two *rpgrip1l*$^{-/-}$ (pink, cyan) adult fish illustrating the 3D spine curvatures in mutants. (**H**) Schematic representation of the two rescue constructs. (**I**) Table presenting the number and phenotype of *rpgrip1l*$^{-/-}$ fish generated for both rescue experiments by stable transgenesis (**J**) Graph representing scoliosis penetrance in transgenic and non-transgenic *rpgrip1l*$^{-/-}$ siblings.

The online version of this article includes the following video, source data, and figure supplement(s) for figure 1:

**Figure supplement 1.** Production and characterization of the *rpgrip1l*$^{Δ}$ and *rpgrip1l*$^{ex4}$ fish lines.

**Figure supplement 1—source data 1.** Raw and unedited rpgrip1l genotyping gel showing wild-type and mutants amplicons from a triplex PCR.

**Figure supplement 1—source data 2.** Edited rpgrip1l genotyping gel showing the cropped region.

**Figure 1—video 1.** Micro-CT of juvenile spines.

https://elifesciences.org/articles/96831/figures#fig1video1

---

## *Rpgrip1l*$^{-/-}$ adult fish display cilia defects along CNS cavities

We observed normal cilia in the neural tube of *rpgrip1l*$^{-/-}$ embryos (*Figure 1—figure supplement 1C*). In 3-month adults, scanning electron microscopy (SEM) at the level of the rhombencephalic ventricle (RhV) showed cilia tufts of multiciliated ependymal cells (MCC) lining the ventricle in controls and straight *rpgrip1l*$^{-/-}$ fish (*Figure 1—figure supplement 1F–G, F'-G'*). In scoliotic *rpgrip1l*$^{-/-}$ fish, cilia tufts were sparse and disorganized (*Figure 1—figure supplement 1H, H'*). Cilia of mono-ciliated ependymal cells were present in *rpgrip1l*$^{-/-}$ fish, some of them with irregular shape. (*Figure 1—figure supplement 1F" and G", H" and H"'*).

## MCC defects at SCO level correlate with scoliosis appearance at juvenile stages

We then took advantage of the asynchrony in scoliosis onset to correlate, at a given stage, cilia defects at different levels of the CNS cavities with the spine curvature status (straight or curved) of *rpgrip1l*$^{-/-}$ juveniles. We analyzed cilia defects in three CNS regions implicated in scoliosis appearance: the spinal cord central canal (since scoliosis develops in the spine), the forebrain choroid plexus (fChP) whose cilia defects have been associated with scoliosis in *katnb1*$^{-/-}$ fish (*Meyer-Miner et al., 2022*) and the subcommissural organ (SCO), which secretes SCO-spondin whose absence leads to scoliosis (*Rose et al., 2020*; *Troutwine et al., 2020*).

In the spinal cord central canal of 8 wpf juvenile fish, cilia density was reduced to the same extent in *rpgrip1l*$^{-/-}$ fish irrespective of the fish curvature status (*Figure 2A–C and E*). In addition, cilia length increased in straight mutants and was more heterogenous than controls in scoliotic mutants (*Figure 2A'–C' and D*). Arl13b content appeared severely reduced in mutants (*Figure 2A"–C"*). In the fChP, mono- and multi-ciliated cell populations were observed with a specific distribution along the anterior-posterior axis, as previously described (*D'Gama et al., 2021*). At the level of the habenula nuclei, cells of dorsal and ventral midline territories were monociliated, while lateral cells presented multicilia tufts (*Figure 2—figure supplement 1K–K"*). *rpgrip1l*$^{-/-}$ juveniles presented an incomplete penetrance of ciliary defects in the fChP, with no correlation between cilia defects and the curvature status (*Figure 2—figure supplement 1L–O"*).

The SCO lies at the dorsal midline of the diencephalic ventricle (DiV), under the posterior commissure (PC) (*Figure 2G and H*, *Figure 2—figure supplement 1A, C*). Cilia in and around the SCO have not been described previously in zebrafish. We showed that, in control animals, Sco-spondin (Sspo) secreting cells varied in number along SCO length, from 24 cells anteriorly to six to eight cells posteriorly (*Figure 2—figure supplement 1A, A', C*) and were monociliated at all anteroposterior levels (*Figure 2G–G'*, *Figure 2—figure supplement 1A'*). The Sspo protein produced from one copy of the transgene Scospondin-GFP$^{ut24}$ (*Troutwine et al., 2020*) formed cytoplasmic granules around SCO cell nuclei, under emerging primary cilia (*Figure 2—figure supplement 1A'*). Conversely, MCCs lateral to the SCO did not secrete Sspo and were prominent towards posterior (*Figure 2H*, *Figure 2—figure supplement 1C*), as observed in the rat brain (*Collins and Woollam, 1979*). Monocilia of Sspo secreting cells were still present but appeared longer in straight *rpgrip1l*$^{-/-}$ juveniles and could not be

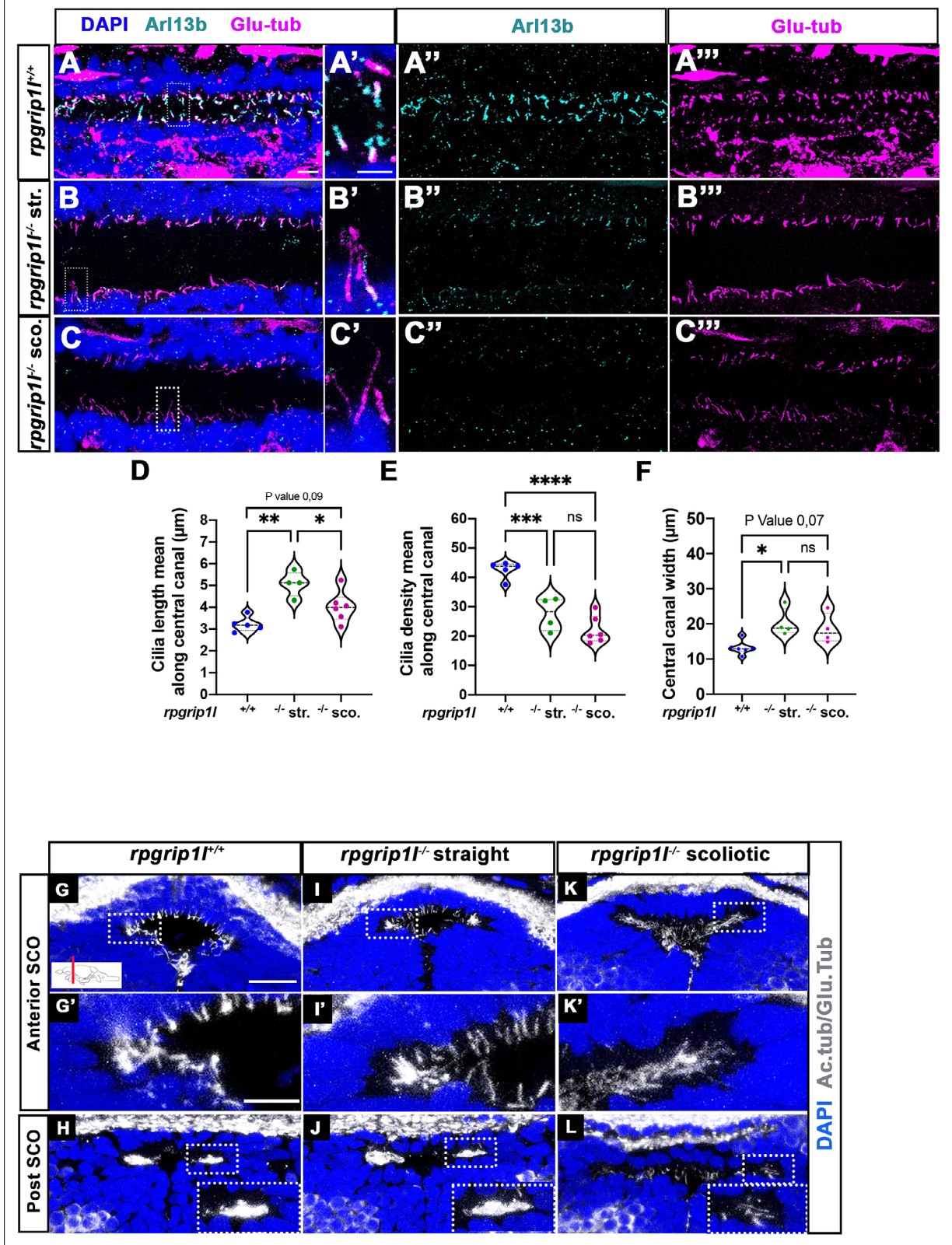

**Figure 2.** Altered ciliogenesis at trunk and subcommissural organ (SCO) levels. (**A-C'''**) Immunostaining of trunk sagittal sections at the level of the spinal cord central canal in *rpgrip1l*+/+ (A-A'''; n=5), straight *rpgrip1l*-/- (B-B'''; n=4) and scoliotic *rpgrip1l*-/- fish (C-C'''; n=6) at 8 weeks (N=2 independent experiments). Cilia were immunostained with Arl13b (cyan, **A'', B'', C''**) and glutamylated-tubulin (magenta, **A''', B''', C'''**). Panels A-C' are merged images with DAPI. Panels A', B' and C' show higher magnification of the squared regions in A, B and C, respectively. Scale bars: 10 µm in A, 5 µm in A'.

*Figure 2 continued on next page*

Figure 2 continued

(D) Violin plot showing the distribution of cilia length along the central canal of *rpgrip1l*[+/+] siblings (n=300 cilia from 5 fish), straight *rpgrip1l*[-/-] (n=267 from 4 fish) and scoliotic *rpgrip1l*[-/-] (n=414 from 6 fish) juvenile fish (N=2). (E) Violin plot showing the distribution of cilia density along the central canal of juvenile *rpgrip1l*[+/+] siblings (n=5), straight *rpgrip1l*[-/-] (n=4) and scoliotic *rpgrip1l*[-/-] fish (n=6) (N=2). Density is the average number of ventral and dorsal cilia lining the CC over a length of 100 µm from three to five sections for each fish. Each dot represents the mean cilia length (D) or cilia density (E) of four to eight sections analyzed at different antero-posterior levels. Statistical analysis was performed using Tukey's multiple comparisons test where * means p-value <0.05; ** means p-value <0.01; *** means p-value <0.001; **** means p-value <0.0001. (F) Violin plot showing the distribution of width of the central canal (nuclear-free region) (in µm) of juvenile *rpgrip1l*[+/+] (n=5), straight *rpgrip1l*[-/-] (n=4) and scoliotic *rpgrip1l*[-/-] fish (n=6) (N=2). Each dot represents the mean of 3 measurements made at different AP levels. Statistical analysis was performed using Tukey's multiple comparisons test where ns means non-significant, * means p-value <0.05. (G–L) Immunostaining of cilia on transverse sections of the brain at SCO level using glutamylated-tubulin and acetylated-tubulin antibodies (both in white), in *rpgrip1l*[+/+] (n=6) (G, G', H), straight *rpgrip1l*[-/-] (n=3) (I, I', J) and scoliotic *rpgrip1l*[-/-] (n=7) (K, K', L) 8 wpf fish (N=2). Nuclei were stained with DAPI (blue). G', I' and K' are higher magnifications of squared regions in G, I and K, respectively. In H, J and L the bottom-right inset represents a higher magnification of the squared region. Scale bars in G and G': 10 µm.

The online version of this article includes the following figure supplement(s) for figure 2:

**Figure supplement 1.** Presence of scospondin aggregates and heterogenous ciliary defects in the fChP of *rpgrip1l*[-/-] juvenile fish.

individualized anymore at anterior SCO level in scoliotic fish: we observed a weaker and denser acetylated and glutamylated tubulin staining that likely corresponded to longer intertwined cilia (*Figure 2I, I', K and K'*). Lateral multicilia tufts were preserved in straight *rpgrip1l*[-/-] juveniles (n=5) (*Figure 2J*) but were missing at posterior SCO level in tail-up or scoliotic *rpgrip1l*[-/-] fish (n=7) (*Figure 2L*).

Thus, *rpgrip1l* mutants develop both monocilia and multicilia defects at juvenile stages, but cilia defects in the spinal cord and in the fChP do not correlate with axis curvature status of the fish. By contrast, the loss of MCC cilia at the SCO strictly correlates with scoliosis onset, suggesting a causal relationship.

## *Rpgrip1l*[-/-] juveniles show ventricular dilations and loss of the Reissner fiber at scoliosis onset

Ciliary beating is an essential actor of CSF flow and of ventricular development in zebrafish larvae (*Kramer-Zucker et al., 2005*; *Olstad et al., 2019*). Thus, ciliary defects in the brain and spinal cord of *rpgrip1l*[-/-] fish could lead to abnormal ventricular and central canal volume and content. In the spinal cord, the lumen of the central canal was enlarged at thoracic and lumbar levels in all *rpgrip1l*[-/-] mutant juveniles, with no correlation to the axis curvature status (*Figure 2A–C, F*). To determine the timing of brain ventricular dilations relative to scoliosis, we analyzed ventricular volume at the onset of spine curvature (*Figure 3A–G*). Ventricular reconstruction was performed on cleared brains of 4 control and 4 *rpgrip1l*[-/-] (3 tail-up and 1 straight) fish at 5 wpf, stained with ZO1 to highlight the ventricular border and with DiI to outline brain shape, focusing on the RhV at the level of the posterior midbrain and hindbrain (*Figure 3A–D*). We identified a significant increase in ventricle volume in *rpgrip1l*[-/-] fish compared to controls, that was restricted to the ventral regions of the RhV in regions ROI4.4 (green sphere in *Figure 3C*) and ROI6 (green sphere in *Figure 3D*), as confirmed by numerical sections (*Figure 3E and G*). Ventricle dilations were also present, although to a lesser extent, in the RhV of the unique straight *rpgrip1l*[-/-] fish of this study (green squares and triangle in *Figure 3F and G*).

Defective ependymal ciliogenesis has been associated with abnormal CSF flow and defective Reissner Fiber (RF) maintenance in several scoliotic zebrafish models (*Grimes et al., 2016*; *Rose et al., 2020*). The RF is mainly composed of Sspo secreted by the SCO and floor plate (FP) and its loss at juvenile stages triggers scoliosis in zebrafish (*Rose et al., 2020*; *Troutwine et al., 2020*). To visualize the RF, we used an antiserum that labels the RF in zebrafish embryos (*Didier et al., 1995*; *Cantaut-Belarif et al., 2018*) and we confirmed the results by introducing one copy of the *sspo-GFP* knock-in allele, sco-spondin-GFP[ut24] (*Troutwine et al., 2020*) by genetic cross into *rpgrip1l*[-/-] animals. The RF formed normally in *rpgrip1l*[-/-] embryos, as expected given the absence of embryonic curvature (*Figure 1—figure supplement 1C*). To study its maintenance in *rpgrip1l*[-/-] juveniles, we immunostained spinal cord longitudinal sections of 7–8 wpf curved or straight fish of the same clutch. The RF was visualized as a 1 µm diameter rod in the central canal of the neural tube and few dots were present in the apical cytoplasm of FP cells (*Figure 3I*) as well as in the SCO of all controls (n=8, *Figure 3L and O*) and in ventricles caudal to the SCO (*Figure 2—figure supplement 1H*). The RF was also present in all *rpgrip1l*[-/-] straight fish (n=9, *Figure 3J*, *Figure 2—figure supplement 1I*). In

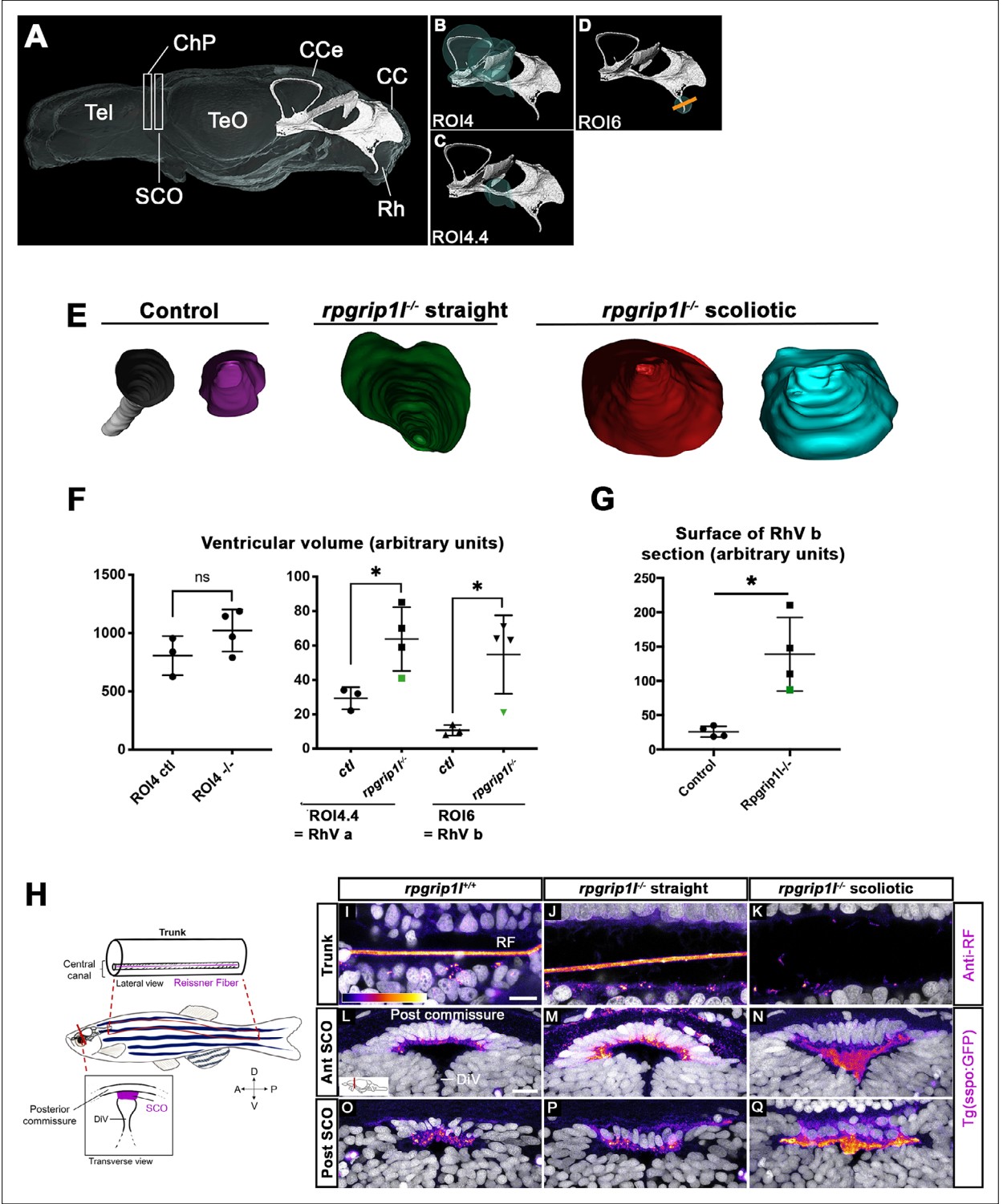

**Figure 3.** *rpgrip1* ⁻ᐟ⁻ juveniles show ventricular dilations and loss of the Reissner Fiber at scoliosis onset. (**A**) Reconstruction of the RhV at the level of posterior midbrain and hindbrain in a transparised 5 wpf control zebrafish brain, stained with ZO1 antibody (ventricular surface) and DiI (global brain shape). Several brain regions are annotated on the reconstruction where Tel means telencephalon, ChP forebrain choroid plexus, SCO subcommissural organ, TeO optic tectum, CCe corpus cerebelli, CC crista cerebellaris and Rh rhombencephalon. (**B–D**) Cyan circles represent the ROIs measured in F. The orange line in D indicates the level of optical sections in E. (**E**) Optical transverse sections showing the caudal part of reconstructed ventricles of a control and two *rpgrip1l⁻ᐟ⁻* fish, one straight and two tail-up. (**F**) Graphs of the ventricle volume at the onset of scoliosis in three control and four *rpgrip1l⁻ᐟ⁻* (one straight and three tail-up) fish. The green dot corresponds to the straight mutant fish. Each dot represents a fish. Statistical analysis was performed

*Figure 3 continued on next page*

Figure 3 continued

using unpaired t-test where ns means non-significant, and * means p-value <0.05. (**G**) Graph of the surface of the optical sections as illustrated in E. The green dot corresponds to a straight mutant fish. Each dot represents a fish. Statistical analysis was performed using unpaired t-test where ns means non-significant, and * means p-value <0.05. (**H**) Schematic representation of a portion of the spinal cord central canal in a lateral view of the fish trunk, showing the Reissner Fiber (RF), and of the SCO in a transverse view of the brain at the level of diencephalic Ventricle (DiV). (**I–Q**) Visualization of the GFP fluorescence in the *sspo-GFP^uts24/+* transgenic line (fire LUT) in sagittal sections of the trunk (**I–K**) (N=3) and transverse sections of the brain at the level of the SCO (N=2) (**L–Q**) in juvenile fish. The corresponding fish are 8 wpf *rpgrip1l^+/+* (**I, L, O**) (n=10); straight *rpgrip1l^-/-* (**J, M, P**) (n=9) and scoliotic *rpgrip1l^-/-* (**K, N, Q**) (n=8). L-N and O-Q show sections at anterior and posterior SCO levels, respectively. Scale bars: 5 µm in I-K, 10 µm in L-Q.

contrast, in *rpgrip1l^-/-* tail-up and scoliotic fish, the RF was absent at all axis levels and a few SCO-spondin-positive debris were present in the central canal (n=8, *Figure 3K*). Transverse sections at the level of the SCO and caudal to the SCO in severely scoliotic fish revealed abnormally packed material in the ventricle (*Figure 3N and Q*, *Figure 2—figure supplement 1J*), while Sspo secreted cells from *rpgrip1l^-/-* at scoliosis onset, that is in juveniles with a slight tail-up phenotype, appeared to produce Sspo-GFP cytoplasmic granules as in controls (*Figure 2—figure supplement 1B, B', D*). We also observed that Sspo aggregates spread anteriorly into the diencephalic ventricle (DiV) at fChP level in scoliotic juveniles (*Figure 2—figure supplement 1G*, n=7/7), a situation never observed in controls (*Figure 2—figure supplement 1E*, n=0/6) or in straight juveniles (*Figure 2—figure supplement 1F*, n=0/5).

Thus, our study uncovers a strict temporal correlation between the loss of MCC at posterior SCO level, impaired RF polymerization and scoliosis onset in *rpgrip1l^-/-* juvenile fish.

## Urp1/2 upregulation does not contribute to scoliosis penetrance or severity in *rpgrip1l^-/-* fish

In scoliotic fish where the RF is not maintained at juvenile stage, *urp1* and *urp2* levels are downregulated (*Rose et al., 2020*). *urp1* and *urp2* encode peptides of the Urotensin 2 family, expressed in ventral spinal CSF-cNs (*Parmentier et al., 2011*; *Quan et al., 2015*). Their expression levels have to be finely tuned to keep a straight axis since their combined loss-of-function induces larval kyphosis that evolves into adult scoliosis (*Bearce et al., 2022*; *Gaillard et al., 2023*), while an increased dose of Urp1/2 peptides induces an upward curvature in embryos (*Zhang et al., 2018*) and juveniles (*Gaillard et al., 2023*; *Lu et al., 2020*). We therefore monitored via qRT-PCR *urp1 and urp2* expression levels on individual straight or scoliotic *rpgrip1l^-/-* juvenile and/or adult fish. *urp2* was the most highly expressed gene among all *urotensin 2* members at juvenile stage (*Figure 4A, D and E* and *Figure 4—figure supplement 1A–E*). *urp2* expression was upregulated in 5 weeks juveniles, both in straight and scoliotic fish and its upregulation was maintained in adult scoliotic fish (*Figure 4A*).

Given the early and long-lasting upregulation of *urp1/2* in *rpgrip1l^-/-* fish, we attempted to rescue axis curvature by decreasing global Urp1/2 production via genetic means by downregulating *urp2* expression level. To that end, we generated double (*rpgrip1l^+/-; urp2^+/-*) heterozygous fish using the recently generated *urp2* deletion allele (*Gaillard et al., 2023*) and crossed them to produce double mutants. Using a curvature quantification method on whole fish (*Figure 4—figure supplement 1I* and Materials and methods), we observed that the removal of one or two *urp2* functional allele(s) in *rpgrip1l^-/-* fish did not lower their curvature index (*Figure 4B*), neither at two mpf (*Figure 4—figure supplement 1F*) nor at four mpf (*Figure 4—figure supplement 1G–H*). We observed that *urp2* mRNA amounts were reduced by removing one or two functional copies of *urp2*, probably as a consequence of mRNA decay (*Gaillard et al., 2023*; *Figure 4D*). We also showed that none of the other *urotensin 2* family members was upregulated by a compensation mechanism upon *urp2* loss (*Figure 4E* for *urp1* and *Figure 4—figure supplement 1C–E* for *urp*, *uts2a* and *uts2b*).

Thus, lowering global Urp1/2 expression level in *rpgrip1l* mutants did not have any beneficial effect on the axis curvature phenotype, ruling out altered *urp1/2* signaling as a main driver of scoliosis in *rpgrip1l* mutants.

## Regulators of motile ciliogenesis, inflammation genes, and annexins are upregulated in *rpgrip1l^-/-* juveniles

To get further insight into the early mechanisms driving spine curvature in *rpgrip1l* mutants, we obtained the transcriptome of the brain and dorsal trunk of 7 *rpgrip1l^-/-* juveniles at scoliosis onset (6

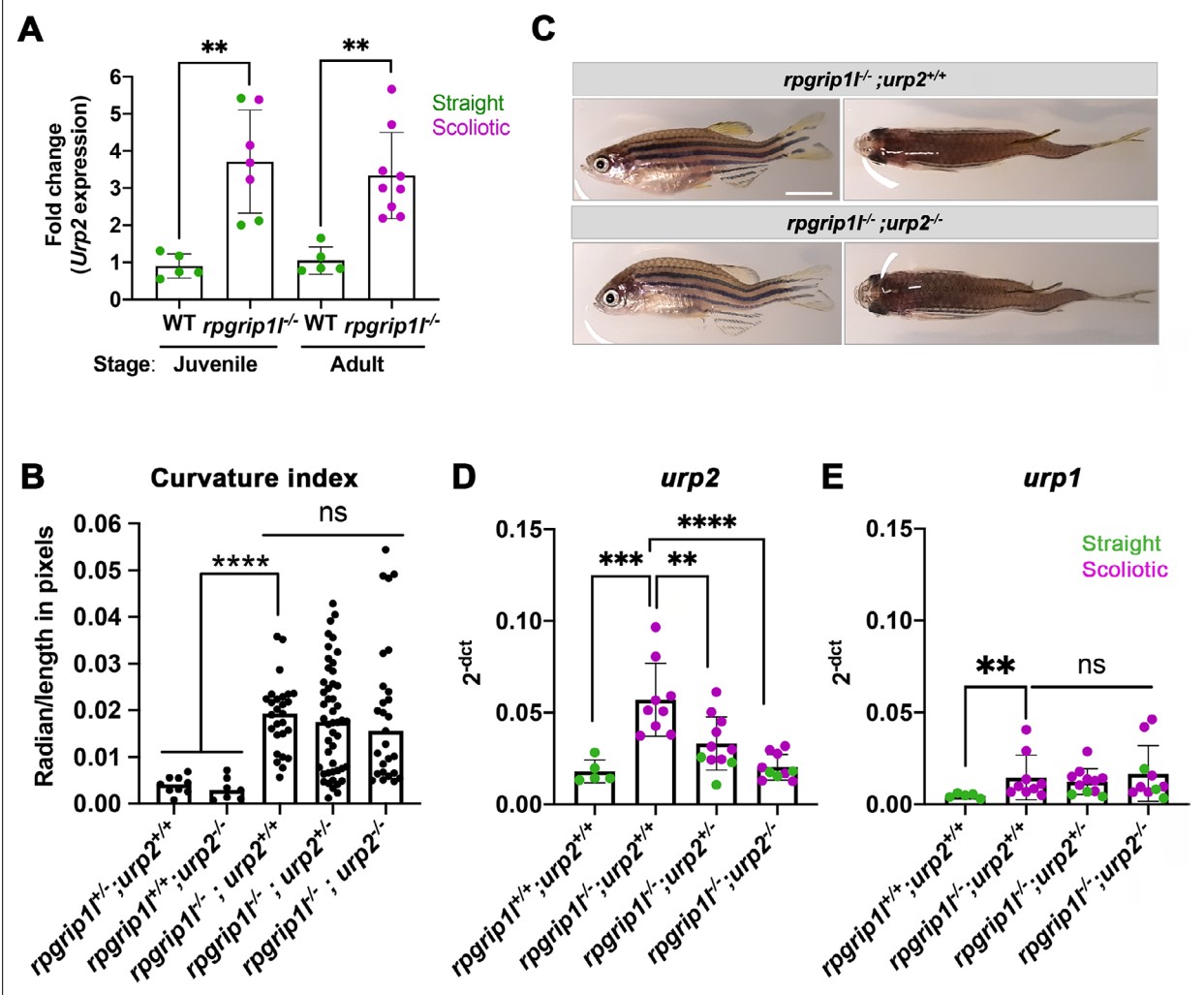

**Figure 4.** Upregulation of *urp1/2* expression does not contribute to axis curvature of *rpgrip1l*−/− fish. Quantitative reverse transcription polymerase chain reaction (qRT-PCR) assaying *urp2* expression level in *rpgrip1l*−/− and *rpgrip1l*+/+ (WT) siblings. Each dot represents the value obtained for one fish. Green and purple dots represent straight and scoliotic *rpgrip1l*−/− fish, respectively. qRT-PCR analysis was performed at 5 wpf (juvenile stage, 5 controls and 7 *rpgrip1l*−/−) and 12 wpf (adult stage, 5 controls and 9 *rpgrip1l*−/−). *urp2* expression levels are increased by approximately 3.5-fold in *rpgrip1l*−/− at both stages. Statistical analysis was performed using unpaired t-test where **p<0.01. Error bars represent s.d. (**B**) Quantification of body axis curvature from dorsal and lateral views in 4 mpf fish issued from [*rpgrip1l*+/−, *urp2*+/−] incrosses. Each point represents the value measured for one fish. Statistical analysis was performed using Tukey's multiple comparisons test where **** p<0.0001 with error bars represent s.d. n=9 [*rpgrip1l*+/+; *urp2*+/+], 7 [*rpgrip1l*+/+; *urp2*−/−], 27 [*rpgrip1l*−/−; *urp2*+/+], 48 [*rpgrip1l*−/−; *urp2*+/−] and 27 [*rpgrip1l*−/−; *urp2*−/−] fish. (**C**) Representative [*rpgrip1l*−/−; *urp2*+/+] and sibling [*rpgrip1l*−/−; *urp2*−/−] fish on lateral and dorsal views at 12 wpf. Scale bar 5 mm. (**D, E**) Expression levels of *urp2* (**D**) and *urp1* (**E**) in adult fish (12 wpf). Each dot represents the value for one fish. Green and purple dots represent straight and scoliotic *rpgrip1l*−/− fish, respectively (n=5 [*rpgrip1l*+/+; *urp2*+/+], 9 [*rpgrip1l*+/+; *urp2*−/−], 11 [*rpgrip1l*−/−; *urp2*+/−] and 10 [*rpgrip1l*−/−; *urp2*−/−]). Each gene expression level is compared to the *lsm12b* housekeeping gene. Statistical analysis was performed using Tukey's multiple comparisons test where ns means non-significant and ** p<0.01, ****; ***p<0.001; p<0.0001. Error bars represent s.d.

The online version of this article includes the following figure supplement(s) for figure 4:

**Figure supplement 1.** Genetic invalidation of *urp2* in *rpgrip1l*−/− fish does not have any beneficial effect on scoliosis penetrance or severity.

tail-up and 1 straight 5 wpf juveniles) and 5 control siblings (2 *rgprip1l*+/+ and 3 *rgprip1l*+/−). Differential gene expression (DGE) analysis was performed with DSEQ2 software within the Galaxy environment. A complete annotated gene list is displayed in *Supplementary file 1* (Trunk) and *Supplementary file 2* (Brain). Filtering the gene list to select for p-adj <0.05 and Log2 FC >0.75 or<−0.75 recovered a vast majority of upregulated genes in the dorsal trunk (75 downregulated vs 614 upregulated genes) and, to a lesser extent, in the brain (1 downregulated, *rpgrip1l*, vs 60 upregulated genes) as illustrated on the volcano plots from DGE analysis of the brain (*Figure 5A*) and trunk (*Figure 5B*). Fifty-four out of 60 genes upregulated in the brain were also upregulated in the trunk, suggesting that, in the

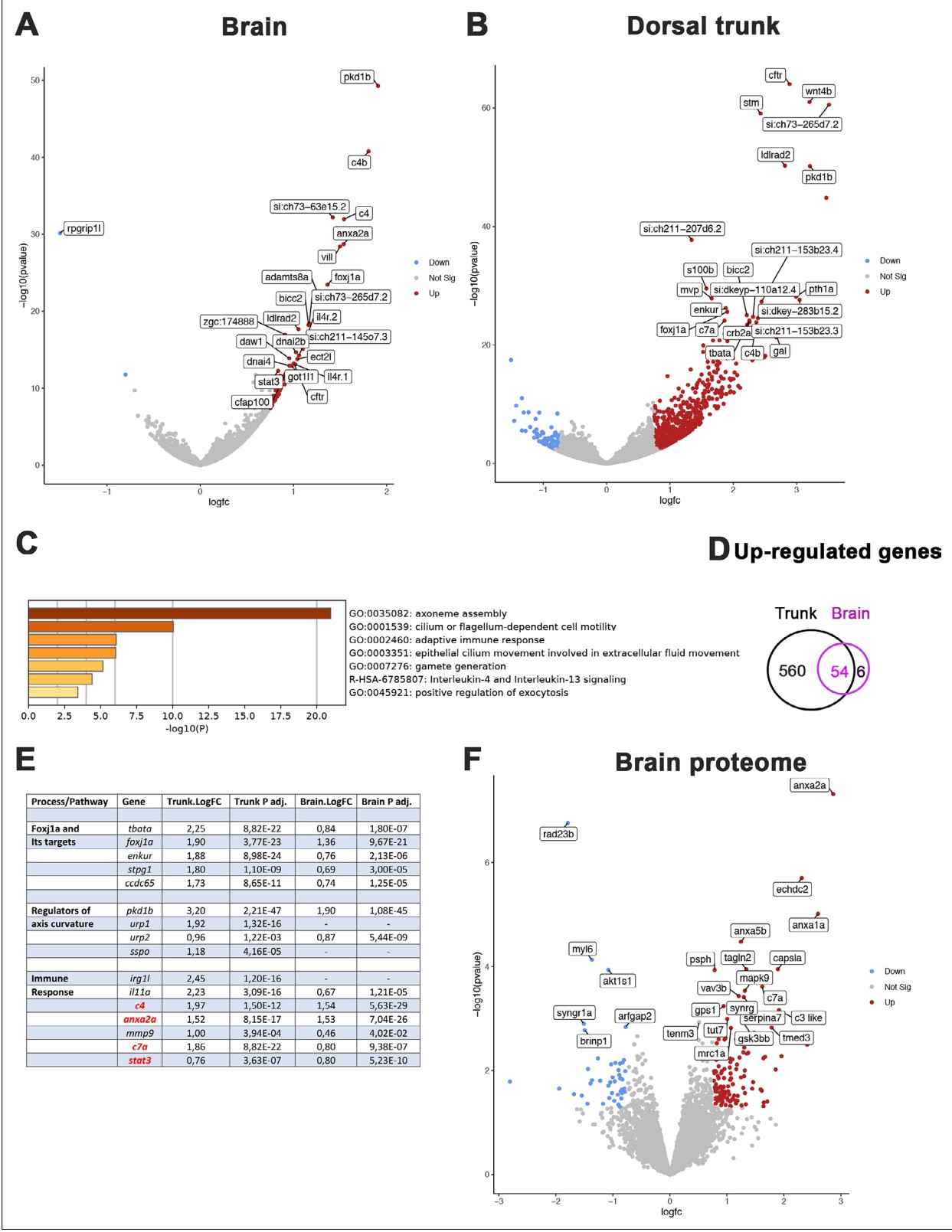

**Figure 5.** Comparative transcriptome and proteome analysis of control and *rpgrip1l-/-* fish. (**A, B**) Volcano-plots showing differentially regulated genes in the transcriptomes of the brain (**A**) and dorsal trunk (**B**) comparing 5 controls (2 wt, 3 *rpgrip1l+/+*) and 7 *rpgrip1l-/-* (6 tail-up and 1 straight) fish at 5 wpf (0.9 cm body length). Each dot corresponds to one gene. A gene was considered deregulated if its associated P-adj. value was inferior to 0.05. Red and blue dots represent up-regulated (Log2FC >0.75), and down-regulated (Log2FC <–0.75) genes, respectively. The names of the top 25 most deregulated

*Figure 5 continued on next page*

*Figure 5 continued*

genes are boxed. (**C**) GO term analysis of biological processes in the brain samples using Metascape. (**D**) Venn diagram of the upregulated genes in brain and trunk samples, showing many common genes between the two regions. (**E**) Selected genes of interest upregulated in the trunk and/or brain of *rpgrip1l*-/- fish. The four genes in red were also found upregulated in the proteome of adult *rpgrip1l*-/- fish. (**F**) Volcano-plot showing differentially expressed proteins in the brain of 5 *rpgrip1l*+/+ fish versus 5 *rpgrip1l*-/- adult scoliotic fish (3 months pf). Each dot corresponds to one protein. A protein was considered differentially expressed if its associated P-adj. value was inferior to 0.05. Red and blue dots represent proteins enriched (Log2FC >0.75) or depleted (Log2FC <–0.75) in mutant brains, respectively. The names of the top 25 most differentially expressed proteins are boxed.

The online version of this article includes the following figure supplement(s) for figure 5:

**Figure supplement 1.** Complementary comparative transcriptome and proteomic analysis of *rpgrip1l*-/- versus control fish.

trunk, these genes were specific to the CNS (*Figure 5D*). In both the brain (*Figure 5C*) and the trunk (*Figure 5—figure supplement 1A*), the most upregulated biological processes in GO term analysis included cilium movement and cilium assembly. Among the 20 most significant GO terms found for upregulated genes in the trunk, 8 were related to ciliary movement/assembly (*Figure 5—figure supplement 1A*). The upregulation in both tissues of *foxj1a,* encoding a master transcriptional activator of genes involved in cilia motility (*Yu et al., 2008*; *Figure 4E*) likely contributed to this enrichment. To confirm this hypothesis, we compared our dataset with those of targets of Foxj1 from *Choksi et al., 2014*; *Mukherjee et al., 2019*; *Quigley and Kintner, 2017*. We found that 106 out of 614 upregulated genes were targets of Foxj1, among which 77 were direct targets (*Supplementary file 3*). Several genes encoding regulators of embryonic axis curvature were also upregulated in *rpgrip1l*-/- fish, among which *pkd1b* (*Mangos et al., 2010*), *urp1* and *urp2* (*Bearce et al., 2022*; *Gaillard et al., 2023*; *Zhang et al., 2018*) and *sspo* (*Cantaut-Belarif et al., 2018*; *Rose et al., 2020*; *Troutwine et al., 2020*; *Figure 5E*). Finally, GO terms enrichment analysis also highlighted processes associated with inflammation and innate and adaptive immune responses, as described for the scoliotic *ptk7* model (*Van Gennip et al., 2018*) sharing for example *irg1l* and complement genes up-regulation (*Figure 5C and E*, and *Figure 5—figure supplement 1A*).

In order to confirm these results and to identify processes affected at the post-transcriptional level in *rpgrip1l* mutants, we performed a quantitative proteomic analysis by mass spectrometry of brains from five *rpgrip1l*-/- adult fish and five control siblings. This analysis detected 5706 proteins present in at least 4 over the 5 replicates, of which 172 are associated with a p-value <0.05 and a Log2FC >+0.75 and<–0.75. 127 proteins were present in higher and 45 in lower quantity, as illustrated on the volcano plot (*Figure 5F* and *Supplementary file 4*). Pathways enriched in *rpgrip1l*-/- brains included regulation of vesicular trafficking and membrane repair processes, as well as proteolysis and catabolic processes, suggesting profound perturbation of cellular homeostasis (*Figure 5—figure supplement 1B*). Among the proteins that were present in highest quantity in mutants, several members of the Annexin family were identified: Anxa2a, Anxa1a, Anxa5b (*Figure 5F* and *Figure 5—figure supplement 1C*). Annexins are associated with membranes from endo and exovesicles as well as plasma membranes and involved in a wide array of intracellular trafficking processes such as cholesterol endocytosis, plasma membrane repair as well as modulation of immune system functions (*Boye and Nylandsted, 2016*; *Grewal et al., 2016*; *Rentero et al., 2018*). Of note, only four proteins of that list, Anxa2a, C4, C7a and Stat3, were the products of genes up-regulated in the transcriptomic study. In agreement with the common enriched GO terms in both studies (*Figure 5C* and *Figure 5—figure supplement 1*), these four proteins are associated with immune response. Overall, *rpgrip1l*-/- mutants at scoliosis onset present deregulated pathways in common with other scoliotic models such as activation of immune response genes, but also specific deregulated pathways such a marked activation of the Foxj1a ciliary motility program and an increase of Annexin levels.

## Astrogliosis precedes scoliosis onset of *rpgrip1l*-/- juvenile fish

Both transcriptomic and proteomic analysis highlighted the occurrence of innate and adaptive immune responses in *rpgrip1l*-/- fish. Several upregulated genes, such as *irg1l, mmp9, and anxa2,* are reported to be expressed in macrophages and/or microglia but also in epithelia or neurons (*Hall et al., 2014*; *Kuhlmann et al., 2014*; *Shan et al., 2015*). We therefore investigated in which cell type(s) Anxa2, the most upregulated protein in the *rpgrip1l*-/- brain proteome, was expressed. Immunostaining of brain sections revealed a strong Anxa2 staining on cells lining brain ventricles, such as SCO cells and ventral ependymal cells of the rhombencephalic ventricle (RhV; *Figure 6B' and D'*) in *rpgrip1l*-/- adult

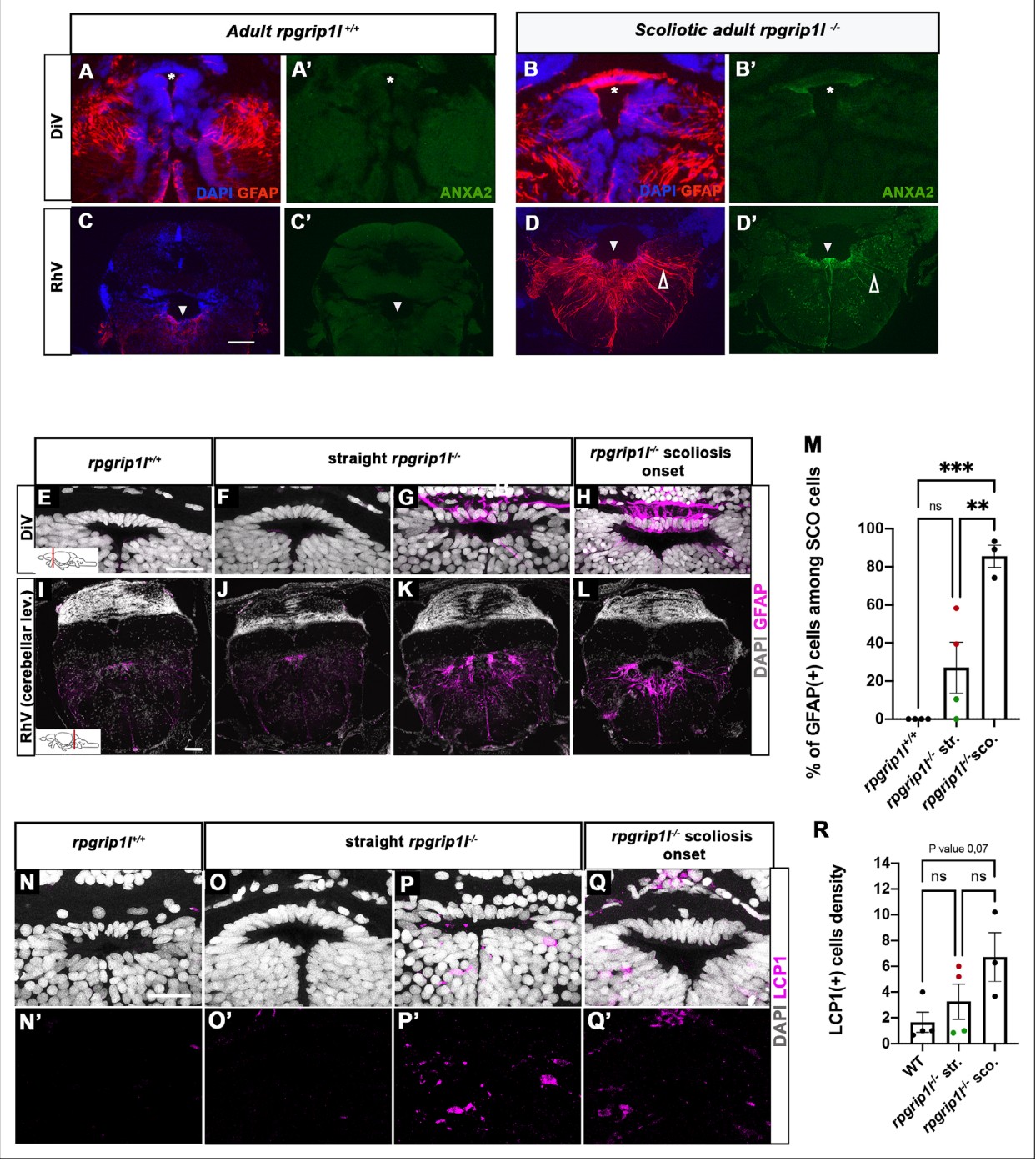

**Figure 6.** Anxa2 and GFAP upregulation and increased number of LCP1-positive cells around CNS ventricles of *rpgrip1l⁻/⁻* juvenile fish at scoliosis onset. (**A-D'**) Immunostaining for GFAP (**A, B, C, D**) and Anxa2 (**A', B', C', D'**) on adult brain transverse sections at DiV (**A-B'**) and RhV at cerebellar level (**C-D'**) levels in controls (**A, A', C, C'**) (n=3) and *rpgrip1l⁻/⁻* scoliotic (**B, B', D, D'**) (n=4) fish. Nuclei are stained with DAPI (blue). White stars indicate the SCO, white triangles point to the ventral RhV region. Open arrows point to a long cellular process co-labeled by Anxa2 and GFAP. (**E–L**) Immunostaining for GFAP (magenta) on transverse sections of the brain at SCO (**E–H**) and RhV (**I–L**) levels in controls (**E, I**) (n=4), straight *rpgrip1l⁻/⁻* (**F, G, J, K**) (n=4) and scoliotic *rpgrip1l⁻/⁻* (**H, L**) (n=3) juvenile (8 wpf) fish. Nuclei are stained with DAPI (white). Scale bar: 125 µM for A-D', 20 µm for E-H and 100 µm for I-L. (**M**) Graph representing the percentage of GFAP-positive cells among the total number of SCO cells in control (n=4), straight *rpgrip1l⁻/⁻* (n=4), and scoliotic *rpgrip1l⁻/⁻* (n=3) fish at scoliosis onset (N=1). Each dot represents the mean ratio of GFAP-positive over -negative cells in 3–6 sections per fish. Green dots correspond to fish with no or very low percentage of GFAP + cells, red dots to fish with a high percentage of GFAP + cells. Statistical analysis was performed with Tukey's multiple comparisons test where ns means non-significant, ** means p-value <0.01 and *** p-value <0.001. Error bars represent s.d. 'str': straight; 'sco': scoliotic. (**N-Q'**) Immunostaining for LCP1 +microglia/macrophages (magenta) on brain transverse sections at SCO levels in a

*Figure 6 continued on next page*

Figure 6 continued

rectangle of 100 μm x 60 μm located under the posterior commissure in controls (n=4) (**N, N'**), straight *rpgrip1l*[-/-] (n=4) (**O-P'**) and *rpgrip1l*[-/-] at scoliosis onset (n=3) (**Q, Q'**) 8 wpf fish (N=1). Nuclei are stained with DAPI (white). Scale bar: 20 μm. (**R**) Quantification of LCP1 + cell density around the SCO in controls (n=4), straight *rpgrip1l*[-/-] (n=4) and scoliotic *rpgrip1l*[-/-] (n=3) at 8 wpf. Twenty-three sections were counted for control, 21 sections for *rpgrip1l*[-/-] straight, and 18 sections for *rpgrip1l*[-/-] scoliotic fish. Each dot represents the mean value for one fish. Green dots correspond to the fish without any GFAP +SCO cells, red dots correspond to the fish with GFAP +SCO cells (SCO cells are defined as the dorsal cells located underneath the posterior commissure and above the diencephalic ventricle). Note that straight *rpgrip1l*[-/-] fish with GFAP +SCO cells present a higher number of LCP1 + cells. Statistical analysis was performed with Tukey's multiple comparisons test where ns means non-significant. Error bars represent s.e.m. 'str':straight 'sco': scoliotic.

The online version of this article includes the following figure supplement(s) for figure 6:

**Figure supplement 1.** Increased GFAP and Annexin 2 staining and LCP1 + cell density in different brain regions.

fish, which was barely detectable in controls (*Figure 6A' and C'*), and to a lesser extent on other brain regions (*Figure 6—figure supplement 1A'-F'*), suggesting a deregulation of ependymal cell homeostasis. In mutants, Anxa2 staining spread over long cellular extensions reaching the pial surface of the rhombencephalon (*Figure 6D'* and *Figure 6—figure supplement 1F'*). This cell shape is characteristic of radial glial cells, a cell type that behaves as neural and glial progenitors and also plays astrocytic functions in zebrafish, as anamniote brains are devoid of stellate astrocytes (*Jurisch-Yaksi et al., 2020*; *Lyons and Talbot, 2015*). To confirm the glial nature of Anxa2-positive cells around brain ventricles in *rpgrip1l*[-/-] fish, we double-labeled brain sections with GFAP, a glial cell marker, and Anxa2. In adult *rpgrip1l*[-/-] fish, SCO cells were strongly double-positive for Anxa2 and GFAP (*Figure 6B and B'*), while controls displayed much weaker labeling for GFAP in the SCO and no labeling for Anxa2 (*Figure 6A and A'*). We made similar observations at rhombencephalic ventricle (RhV) and other brain levels: ependymal cells were positive for Anxa2 and strongly positive for GFAP, and GFAP labeling was much higher in mutants than in controls (*Figure 6C–D'*; *Figure 6—figure supplement 1A–F'*). GFAP overexpression is a hallmark of astrogliosis, a process in which astroglial cells react to the perturbed environment, as shown during brain infection, injury, ischemia or neurodegeneration (*Escartin et al., 2021*; *Matusova et al., 2023*; *Zamanian et al., 2012*). Thus, the strong increase of GFAP staining combined with Anxa2 overexpression in brain ependymal cells of *rpgrip1l*[-/-] adult fish strongly suggests that these cells undergo astrogliosis as observed in other animal models following CNS injury (*Chen et al., 2017*; *Zamanian et al., 2012*).

To assay whether an astrogliosis-like phenotype was already present at scoliotic onset, we immunostained GFAP on juvenile brain sections of straight and tail-up mutants and controls. Tail-up juveniles (n=3/3) presented a strong GFAP staining in almost all cells of the SCO and along the ventral rhombencephalic ventricle, higher than control levels (*Figure 6E, I, H, L and M*) while the phenotype of straight fish was heterogeneous: two out of four juveniles presented GFAP-positive cells (*Figure 6G and K* and red dots on *Figure 6M* graph), while the two other did not (*Figure 6J* and green dots on *Figure 6M* graph) as confirmed by cell quantification (*Figure 6M*). As scoliosis onset is asynchronous in *rpgrip1l*[-/-], heterogenous GFAP staining among straight mutant fish may support an early role of astrogliosis at SCO and RhV levels in scoliosis onset.

## Immune cell enrichment around the SCO and within tectum parenchyma is detected prior to scoliosis onset

Astrogliosis may be reinforced or induced by an inflammatory environment containing cytokines released by microglia or macrophages or the presence of high levels of ROS and NO (*Sofroniew, 2015*). We thus performed immunostaining for the LCP1/L-Plastin marker, which labels both macrophages and microglia (*Figure 6N–Q'*), in adjacent sections to those used for GFAP (*Figure 6E–L*). We found that the scoliotic fish presented a higher number of LCP1-positive cells of ameboid shape within SCO cells (*Figure 6Q, Q' and R*). Strikingly, the straight mutants (2/4) that presented a high GFAP staining within SCO cells (*Figure 6G and K*) also showed a higher number of LCP1-positive cells around the SCO (*Figure 6P and P'*; red dots in *Figure 6R*) than in controls (*Figure 6N and N'*), while the other straight mutants did not (*Figure 6O and O'*, green dots in *Figure 6M*) suggesting positive regulation between astrogliosis and macrophages/microglia recruitment. LCP1 + cells were also detected in other regions of the brain (*Figure 6—figure supplement 1G–K*). In the optic tectum, where resident LCP1-positive macrophages/microglia were present in controls, a twofold increase in

the number of ramified LCP1$^+$ cells was observed in straight *rpgrip1l*$^{-/-}$ juveniles compared to controls (*Figure 6—figure supplement 1H–J*). Surprisingly, scoliotic *rpgrip1l*$^{-/-}$ fish showed similar LCP1$^+$ cell number as control fish in the tectum (*Figure 6—figure supplement 1G*), suggesting a transient increase of macrophage/microglia activation preceding scoliosis.

Our data thus uncovers an early astrogliosis process initiated asynchronously at the SCO and RhV levels of straight *rpgrip1l*$^{-/-}$ fish, favored by a transient inflammatory state, and preceding the loss of multiciliated tufts around the SCO and that of the RF, that we only detected in curved *rpgrip1l*$^{-/-}$ fish. Astrogliosis then spreads along the brain ventricles and becomes prominent in brain of scoliotic fish. In addition, we found among the upregulated genes in dorsal trunk at scoliosis onset, a number of activated astrocyte markers (*Matusova et al., 2023*) such as *s100b*, *c4b*, *gfap*, *glast/slc1a3b*, *viml*, *ctsb*, (*Figure 5—figure supplement 1D*), showing that astrogliosis spread over the whole central nervous system.

## Brain neuroinflammation and astrogliosis contribute to scoliosis penetrance and severity in *rpgrip1l*$^{-/-}$ fish

Our data suggest that astrogliosis and neuro-inflammation could play a role in triggering scoliosis in *rpgrip1l* mutants. We thus aimed to counteract inflammation before curvature initiation by performing drug treatment with the hope of reducing scoliosis penetrance and severity. We used the NACET anti-oxidant and anti-inflammatory drug that successfully reduced scoliosis of *sspo* hypomorphic mutant fish (*Rose et al., 2020*). We first assayed the biological activity of our NACET batch on the ciliary motility mutant *dnaaf1*$^{-/-}$ (*Sullivan-Brown et al., 2008*). Indeed, NACET was shown to suppress the embryonic tail-down phenotype of the ciliary motility mutant *ccdc151*$^{ts}$ (*Rose et al., 2020*). We found that 3 mM NACET treatment suppressed the *dnaaf1*$^{-/-}$ tail-down phenotype (*Figure 7—figure supplement 1A–F*). To suppress scoliosis, we treated the progeny of *rpgrip1l*$^{+/-}$ incrosses from 4 to 12 wpf with 1.5 mM NACET as in *Rose et al., 2020*. This long-term treatment reduced scoliosis penetrance from 92% to 58% and markedly reduced spinal curvature index (*Figure 7A–C*), measured and illustrated as shown in *Figure 4—figure supplement 1I*.

We then asked whether NACET treatment suppressed astrogliosis and neuroinflammation in *rpgrip1l*$^{-/-}$ treated fish. Indeed, the intensity and number of GFAP + cells within the SCO were reduced in slightly curved fish and alleviated in straight fish (*Figure 7E and F*). Furthermore, the quantification of Lcp1-positive cells in the SCO region showed a drastic reduction of these cells in NACET-treated fish, back to levels found in *rpgrip1l*$^{+/+}$ fish (*Figure 7D and E*).

Our data thus show that *rpgrip1l*$^{-/-}$ juvenile fish display brain astrogliosis and inflammation at scoliosis onset and that an anti-inflammatory/anti-oxidant treatment of *rpgrip1l*$^{-/-}$ juveniles is able to maintain a straight axis in approximately one third of the mutants and to slow down curve progression in the other two thirds. The beneficial effect of NACET on axis curvature correlates with the combined reduction of astrogliosis and immune cell density.

## Brain astrogliosis is present in zebrafish *cep290*$^{fh297/fh297}$ scoliotic fish

To investigate whether abnormal SCO cilia function and brain astrogliosis may be a general mechanism of scoliosis development in transition zone mutants, we studied the *cep290*$^{fh297/fh297}$ zebrafish mutant. Zebrafish *cep290* mutants display variable axis curvature defects at embryonic stages as well as slow retinal degeneration, and develop juvenile scoliosis (*Lessieur et al., 2019*; *Wang et al., 2022*). Like *RPGRIP1L*, the human *CEP290* gene encodes a TZ protein and its mutation leads to severe neurodevelopmental ciliopathies (*Baala et al., 2007*). All *cep290*$^{fh297/fh297}$ juveniles developed a fully penetrant scoliosis by 4 wpf in our fish facility (n: 30/30). We analyzed cilia presence within the SCO by immunostaining on sections (*Figure 8A–F'*). At 4 wpf, the SCO had not reached its final adult width (14 vs 24 cells wide in the anterior SCO), and each of its cells harbored a short monocilium labeled by glutamylated tubulin (*Figure 8A–C*). Multicilia tufts on SCO lateral walls were not as clearly visible as in adults (*Figure 2F*) but patches of glutamylated tubulin staining were detected on both sides of the posterior SCO (*Figure 8B–C'*). In *cep290*$^{-/-}$, monocilia of the SCO appeared longer than in controls (N=3/3) and pointed towards an abnormally elongated diencephalic ventricle (*Figure 8D–F*). Lateral patches of glutamylated tubulin staining were present in *cep290*$^{-/-}$ posterior SCO as in controls (*Figure 8E–F'*). Thus, cilia morphology is perturbed within *cep290*$^{-/-}$ SCO monociliated cells, as well as the overall morphology of the diencephalic ventricle at this level.

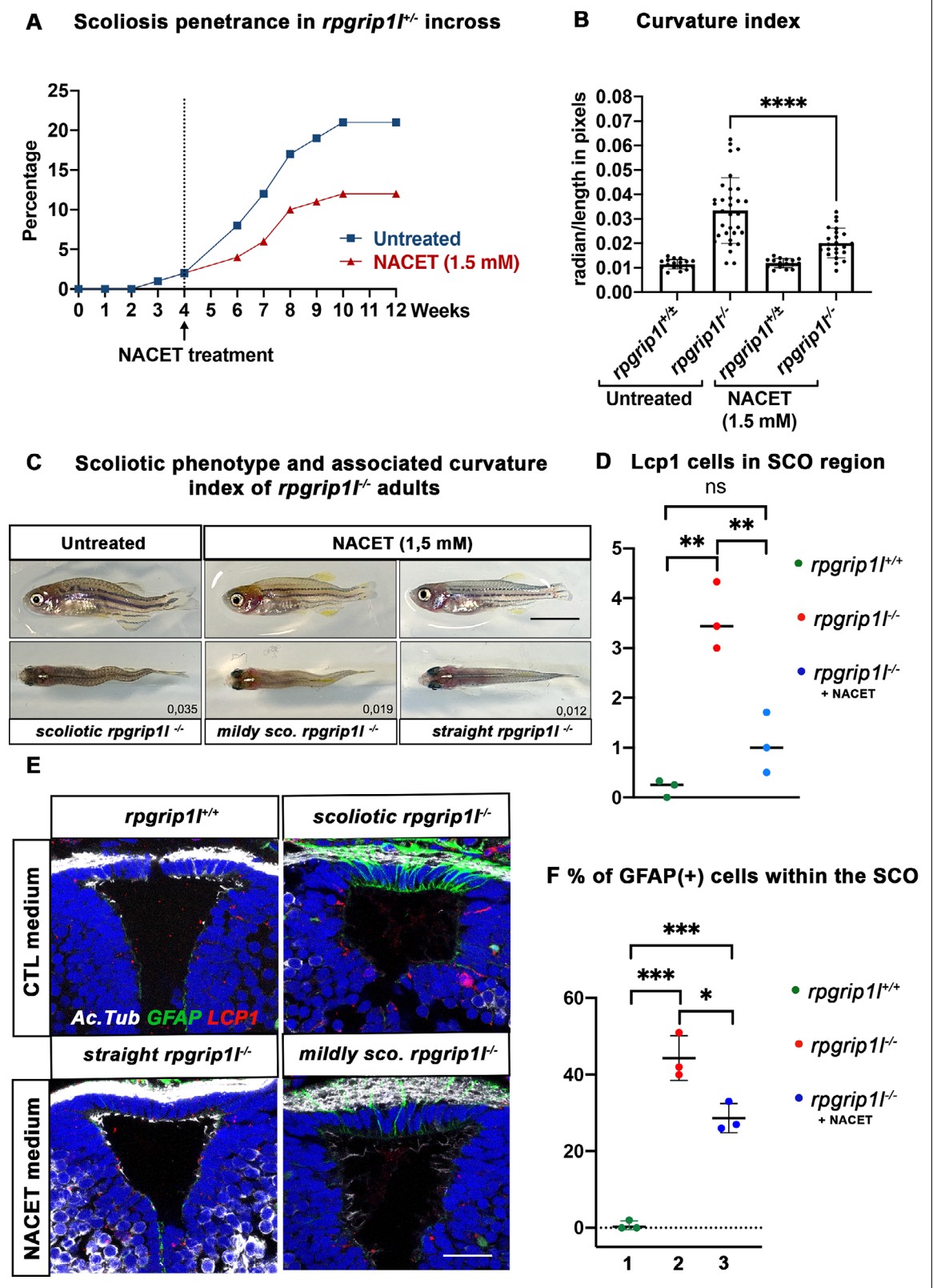

**Figure 7.** NACET treatment reduces scoliosis penetrance and severity in *rpgrip1l⁻/⁻* and decreases astrogliosis and LCP1 + cell number at SCO level. (**A**) Kinetics of scoliosis development in *rpgrip1l⁺/⁻* incrosses in the presence (n=150 fish) or absence (n=174 fish) of NACET treatment (1.5 mM) from 4 to 12 weeks (N=1). 'str': straight; 'sco': scoliotic. (**B**) Quantification of body axis curvature from dorsal and lateral views at 13 wpf, at the end of the NACET treatment. Each dot corresponds to one fish. Among the 174 fish raised in untreated condition, we measured 18 controls (*rpgrip1l⁺/⁻*) and 33 *rpgrip1l⁻/⁻*,

*Figure 7 continued*

and upon NACET treatment, 16 controls (*rpgrip1l*[±/+]) and 23 *rpgrip1l*[-/-]. *rpgrip1l*[±/-] represents a pool of *rpgrip1l*[+/+] and *rpgrip1l*[+/-]. Statistical analysis was performed using Tukey's multiple comparisons test where * means p-value <0.05; ** means p-value <0.01; *** means p-value <0.001; **** means p-value <0.0001. Error bars represent s.d. (**C**) Representative pictures of *rpgrip1l*[-/-] fish with their curvature index without or after NACET treatment, at 13 wpf. Scale bar is 0.5 cm. Left panels correspond to an untreated *rpgrip1l*[-/-] scoliotic fish on lateral (upper) and dorsal (lower) views. Middle panels represent treated *rpgrip1l*[-/-], with a lower curvature index than scoliotic fish on lateral (upper) and dorsal (lower) views and right panels are pictures of rescued straight *rpgrip1l*[-/-] fish on lateral (upper) and dorsal (lower) views. (**D**) Graph representing the number of LCP1 + cells in wt (n=3), untreated *rpgrip1l*[-/-] (n=3) and NACET-treated *rgprip1l*[-/-] (n=3) fish in the SCO region (N=1). (**E**) Immunostaining for GFAP (green), Acetylated tubulin (white) and LCP1 (red) on adult brain transverse sections at SCO level in untreated (upper panels) and NACET-treated (lower panels) fish. Nuclei are stained with DAPI (blue). No GFAP staining was present in controls (n=3) (upper left), while untreated mutants displayed strong GFAP labeling (upper right) (n=3). NACET treatment totally suppressed gliosis in a straight *rpgrip1l*[-/-] fish (lower right panel) (n=1) and diminished the intensity of GFAP labeling in slightly tail-up *rpgrip1l*[-/-] (n=2). Scale bar: 12.5 µm. (**F**) Graph presenting the percentage of GFAP positive cells among the total number of SCO secretory cells compared between controls (n=3), untreated *rpgrip1l*[-/-] (n=3), and NACET treated *rpgrip1l*[-/-] adult fish (n=3) (N=1). Each dot represents the mean of % of GFAP-positive cells present in 3–6 sections per fish.

The online version of this article includes the following figure supplement(s) for figure 7:

**Figure supplement 1.** Biological assay of NACET activity on *dnaaf1*[-/-] embryos.

To test for the presence of astrogliosis, we quantified the number of SCO cells presenting high GFAP labeling on several sections along the A/P extent of the SCO in three control and three *cep290*[-/-] brains (***Figure 8G–K***). Fifty percent of SCO cells presented strong GFAP staining in fully scoliotic juveniles, which was not detectable in controls (***Figure 8H and K***, N=2/3), while a lower proportion (22%) of SCO cells was GFAP positive in the juvenile with a milder curvature (***Figure 8M*** N=1/3). A strong GFAP staining was also detected in some diencephalic radial glial cells (***Figure 8K***). On the same sections, immune cells were quantified by LCP1 staining. A mean of 4 LCP1-positive cells per SCO field was found in *cep290*[-/-], versus 1 in controls (***Figure 8I–L and N***). As in *rpgrip1l* mutants, both abnormal monocilia, astrogliosis and immune cell enrichment are evidenced at SCO levels in *Cep290*[-/-] (3/3). Furthermore, astrogliosis was observed within radial glial cells lining the RhV at cerebellar level (***Figure 8P–R***) and at anterior hindbrain level (***Figure 8T–V***), as shown in *rpgrip1l*[-/-] scoliotic brain (***Figure 6D, K and L***). This indicates that astrogliosis is shared by these two zebrafish scoliotic models, even though they develop axis curvature at different stages (3–4 wpf for *cep290*[-/-] versus 5–12 wpf for *rpgrip1l*[-/-]).

## Discussion

In this paper, we dissected the mechanisms of scoliosis appearance in a zebrafish mutant for the TZ gene *rpgrip1l*, taking advantage of its asynchronous onset to decipher the chronology of events leading to spine torsion. Transcriptome analysis of the brain and trunk at scoliosis onset in *rpgrip1l*[-/-] fish compared to control siblings revealed increased expression of *urp1/2*, *foxj1a* and its target genes, as well as genes linked to immune response. Genetic experiments ruled out an involvement of increased Urp1/2 signaling in scoliosis development. Cilia defects appeared asynchronously in different CNS regions, while RF loss strictly coincided with scoliosis onset. Strikingly, we observed a strong increase of GFAP staining along the diencephalic and rhombencephalic ventricles, indicative of astrogliosis. This astrogliosis preceded scoliosis onset and then spread posteriorly and increased in severity. In the region of the SCO, the appearance of astrogliosis and neuroinflammation coincided temporally, preceding the onset of RF depolymerization and spine torsion. NACET treatment showed that increased astrogliosis and inflammation participated in the penetrance and severity of scoliosis in *rpgrip1l*[-/-] fish. Finally, we showed that another TZ gene mutant, *cep290*, also displayed astrogliosis at the time of scoliosis onset. Thus, the cascade of events uncovered in the *rpgrip1l* mutant sheds a new light on the origin of zebrafish scoliosis caused by ciliary transition zone defects.

Cilia motility defects lead to RF loss and to scoliosis in several zebrafish mutants (***Grimes et al., 2016***; ***Rose et al., 2020***). *Rpgrip1l* encodes a TZ protein important for cilia integrity and function in many eukaryotic organisms (***Gogendeau et al., 2020***; ***Jensen et al., 2015***; ***Wiegering et al., 2018***). Our data show that *rpgrip1l* deficiency in zebrafish affects cilia maintenance differentially depending

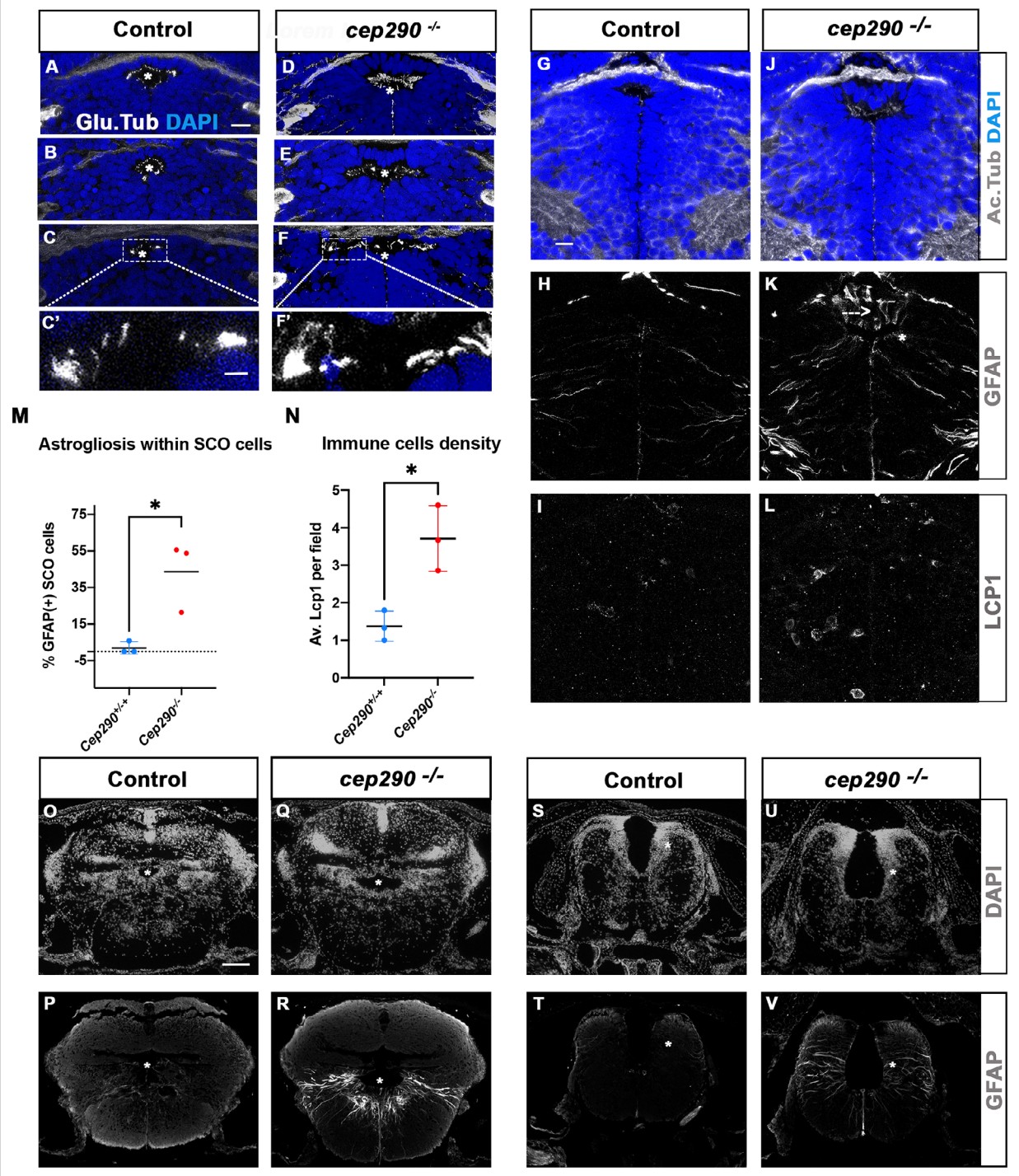

**Figure 8.** Altered ciliogenesis, GFAP upregulation and increased number of LCP1-positive cells around ventricles of *cep290*[-/-] juvenile fish at 4 wpf. (**A-F'**) Immunostaining of the cilia marker glutamylated tubulin (white) on 4 wpf brain transverse sections at 3 antero-posterior SCO levels in 4 wpf control (A-C', n=3) and *cep290*[-/-] scoliotic (D-F', n=3) fish. Nuclei are stained with DAPI (blue). White stars indicate the diencephalic ventricle (DiV), above which lies the SCO. White rectangles in C and F delineate the close-up on cilia (**C', F'**). Scale bar: 10 µm for A-F, 2.5 µm for C'-F'. (**G–L**) Immunostaining for Acetylated Tubulin (**G, J**), GFAP (**H, K**) and LCP1 (**I, L**) on transverse sections of the brain at SCO level in controls (**G–I**) (n=3), and *cep290*[-/-] (**J–L**) (n=3) juvenile (4 wpf) fish. In G and J, nuclei are stained with DAPI (blue). Scale bar: identical to A-F. (**M**) Graph representing the percentage of GFAP-positive cells among the total number of SCO secretory cells in control (n=3) and scoliotic *cep290*[-/-] (n=3) fish at scoliosis onset (N=1). Each dot represents the mean ratio of GFAP-positive over total SCO secretory cells in 5–7 sections per fish. (**N**) Quantification of LCP1 + cell density around the SCO in controls (n=3) and curved *cep290*[-/-] (n=3) fish at 4 wpf. 16 sections were counted for control, 18 sections for *cep290*[-/-] scoliotic fish. Each dot represents the mean

*Figure 8 continued on next page*

*Figure 8 continued*

value for one fish. Statistical analysis was performed with unpaired T test where * means p-value < 0.05. Error bars represent s.d. (**O–V**) Immunostaining for GFAP (white) on brain transverse sections at cerebellum levels (**P, R**) and anterior hindbrain level (**T, V**) in control (**P, T**) (n=3), and *cep290*[-/-] (**R, V**) (n=3) fish. Corresponding nuclear staining (DAPI in white) is shown above (**O, Q, S, U**). Scale bar: 125 μm.

on the CNS territory and the developmental stage, raising the question of which ciliated cells are responsible for RF loss and scoliosis initiation in this mutant. The full rescue observed after re-introducing RPGRIP1L in *foxj1a*-expressing cells points towards a crucial function of the protein in cells lining CNS cavities. Foxj1a-positive cell populations encompass both monociliated and multiciliated ependymal cells, radial glial progenitors, glial cells of the SCO and the ChP, as well as CSF-contacting sensory neurons in the spinal cord (*D'Gama et al., 2021*; *Prendergast et al., 2023*; *Ribeiro et al., 2017*; *Van Gennip et al., 2018*). Recent work suggested that impairment of ciliogenesis in the fChP of the *katnb1* mutant could play a role in inducing axis curvature (*Meyer-Miner et al., 2022*). In the case of the *rpgrip1l* mutant, we temporally uncorrelated fChP cilia defects from scoliosis onset. This, combined with the fact that the total loss of multicilia in the fChP of (*foxj1b*[-/-], *gmnc*[-/-]) double mutants does not trigger axis curvature (*D'Gama et al., 2021*) suggests that another *foxj1a*-expressing population is crucial for axis straightness. Our results strongly suggest that cilia defects in the SCO trigger axis curvature in *rpgrip1l* mutants. Two cilia populations were affected in the SCO of *rpgrip1l* mutant juveniles at the onset of scoliosis: cilia of monociliated glial cells that secrete SCO-spondin, which appeared longer in mutants than in controls, and multiciliated tufts at SCO exit, which were lost in scoliotic mutants. Both defects could be responsible for the phenotype, by perturbing CSF flow and content (presence of SCO-spondin aggregates in ventricles;left side on the *Supplementary file 5*) and leading to ventricular dilations or by perturbing primary cilia signaling within SCO cells (right side of the *Supplementary file 5*). The observation that multiciliated cells were not yet fully differentiated in *cep290* mutants at scoliosis onset is in favor of a prominent role of monocilia. This is consistent with the literature showing that most ciliary mutant models develop axis curvature at 3–4 weeks, a stage when brain multiciliated cells are not yet differentiated (*D'Gama et al., 2021*). Still, as scoliosis onset in *rpgrip1l* mutant is asynchronous from 5 to 12 weeks, MCC loss at SCO exit could contribute to defective RF polymerization.

Our model for scoliosis appearance in the rpgrip1l mutant is presented in the *Supplementary file 5*. Since we have observed several straight *rpgrip1l*[-/-] with astrogliosis and increased immune cells presence around the SCO, we propose that *rpgrip1l*[-/-] SCO and astroglial cells experience an altered signalling that activates immune cells recruitment, producing an inflammatory environment. This would promote multicilia loss as a secondary event (*Supplementary file 5* - Graphical abstract, orange arrow; *Lattke et al., 2012*) leading to RF depolymerization and spine curvature. However, as *sspo*[-/-] hypomorphic mutants which present aggregates within ventricles have been shown to present an inflammatory response (*Rose et al., 2020*), it is equally possible that SCO astrogliosis is a downstream consequence of the presence of these proteoglycans aggregates in contact with primary cilia or ventricular cell surfaces (*Supplementary file 5* - Graphical abstract, green arrow). These two scenarios may also act in parallel to reinforce scoliosis onset and ependymal cilia loss. Future experiments will help to discriminate between these two scenarios.

Our transcriptome analysis and qRT-PCR data challenges a model proposed for axis straightness of zebrafish embryos in which the loss of RF leads to a down-regulation of *urp2* and *urp1* expression (*Zhang et al., 2018*) since we observed an upregulation of both genes before and after RF loss. As forced expression of *urp1/2* induces a tail-up phenotype in embryos and juveniles (*Zhang et al., 2018*; *Gaillard et al., 2023*; *Lu et al., 2020*), we attempted to rescue *rpgrip1l*[-/-] axis curvature by down-regulating *urp1/2* expression. No beneficial effect on scoliosis penetrance or severity was observed, indicating that increased *urp1/2* expression level does not significantly contribute to axis curvature in *rpgrip1l*[-/-] fish.

In this study, we found an unexpected and strong defect characterized by enhanced GFAP and Anxa2 expression, which we identified as astrogliosis (also called reactive astrogliosis). Astrogliosis is a reaction of astroglial cells to perturbed homeostasis of the CNS, characterized by molecular and phenotypic changes in these cells (*Escartin et al., 2021*; *Matusova et al., 2023*). In zebrafish,

radial glial cells are endowed with both neurogenic and astrocytic functions, and are thus also called astroglial cells (*Jurisch-Yaksi et al., 2020*). Astrogliosis in *rpgrip1l* mutants arose in a subdomain of Foxj1a-positive cells, within the DiV and ventral ependymal cells lining the RhV. The observation of astrogliosis in some straight juvenile mutants suggests an early role of this defect in scoliosis onset.

Astrogliosis along the ventricular cavities could arise in response to local mechanical or chemical perturbations caused by cilia motility defects, abnormal CSF flow and content and ventricular dilations. In two murine models of primary ciliary dyskinesia, decreased CSF flow was associated with gliosis at juvenile stage (*Finn et al., 2014*). A response to CSF pertubation in *rpgrip1l* mutants is also suggested by the strong upregulation of Foxj1a and its target genes. *foxj1a* expression is also reported to be upregulated in the CNS of three other scoliotic models in addition to *rpgrip1l*, namely in *ptk7, sspo* and *katnb1* mutants (*Escartin et al., 2021*; *Meyer-Miner et al., 2022*; *Van Gennip et al., 2018*), suggesting that it constitutes a common response to similar ventricular and CSF defects. Interestingly, *foxj1a* expression is also upregulated upon zebrafish embryonic (*Cavone et al., 2021*; *Hellman et al., 2010*) or adult (*Ribeiro et al., 2017*) spinal cord injury. The up-regulation in the *rpgrip1l* mutant proteome of the membrane repair module Anxa2-S100a10-Ahnak (*Bharadwaj et al., 2021*; *Figure 4C*) together with the Foxj1-induced motility module detected in the transcriptomic analysis may indicate ongoing reparation attempts of CNS ventricles and associated neural tissues.

Moreover, the presence of few Lcp1 + cells around the mutant SCO suggests that microglia/macrophages could participate in astrogliosis establishment and reinforcement. This dialogue between immune Lcp1 + cells and astroglial cells is evidenced during early stages of regeneration after acute injury or cellular damages (*Cavone et al., 2021*). In our bulk RNAseq analysis performed at scoliosis onset, we have indications of the upregulation of microglia and macrophage markers (*p2ry12* and *mpeg1*) at trunk levels and of numerous markers of activated astrocytes characterized in several mammalian astrogliosis models (*Matusova et al., 2023*). A similar dialogue is detected in polycystic kidney disease models where compromised primary cilia signalling leads to uncontrolled cytokines secretion by epithelial cells, a situation that favors local immune cells recruitment and proliferation (*Viau et al., 2018*).

What are the intracellular mechanisms involved in astrogliosis induction in ciliated ventricular cells? Candidate pathways emerge from our multi-omics studies. Proteomic data showed a 2.4-fold increase in GSK3bb amount in *rpgrip1l*[-/-] mutant brains (p-value <0.001, *Supplementary file 3*). In several neurodegenerative murine animal models, GSK3b enzymatic activity was shown to promote inflammation and gliosis (*Jorge-Torres et al., 2018*; *Medina and Avila, 2010*; *Mines et al., 2011*). Another potential trigger of astrogliosis may be a defective mitochondrial activity as the amounts of four proteins involved in electron transport chain activity (*gpd1b, mt-nd6, pdia5, and dmgdh*) were reduced by 40% to 86% in the analysis of mutant brain proteome (*Supplementary file 4*). We think this is of particular interest in light of a recent report demonstrating that impaired mitochondrial activity within ciliated astrocytes leads to the induction of the Foxj1 ciliary motility program, the elongation and distortion of astrocyte primary cilia as well as reactive astrogliosis (*Ignatenko et al., 2023*), three phenotypes also observed in *rpgrip1l*[-/-] brains.

Finally, our observation of widespread astrogliosis in two TZ gene mutants, *rpgrip1l* and *cep290*, suggests that it could represent a general mechanism involved in scoliosis downstream of cilia dysfunction. Of note, astrogliosis markers such as *glast/slc1a3b*, *vim*, *s100b*, *c4*, *timp2b*, *gfap* and *ctssb.2*, are also upregulated in the transcriptome of *ptk7 mutants* (*Van Gennip et al., 2018*). We propose that sustained astrogliosis might impair neuronal survival and activity, crucial for proper interoception and locomotor control, thus leading to axis curvature in the context of a rapidly growing axial skeleton. A similar context of astrocytosis associated with spinal cord dilation has been observed in human patients with syringomyelia. This population with a high incidence of scoliosis can present hypersignals on T2 MRI scans, which were proposed to reflect local astrocytosis and could be validated by biopsies in rare cases (*Sherman et al., 1987*). Further imaging studies will need to be performed to validate a potential link between developing astrogliosis and the onset and progression of idiopathic scoliosis in humans.

# Materials and methods

## Key resources table

| Reagent type (species) or resource | Designation | Source or reference | Identifiers | Additional information |
|---|---|---|---|---|
| Strain (*Danio rerio*) | Zebrafish wild-type AB or (TL x AB) hybrid strains | IBPS aquatic facility, Paris | N/A | |
| Strain (*Danio rerio*) | Zebrafish *rpgrip1l*$^{\Delta}$ mutant strain | This manuscript, Zfin: *rpgrip1l*$^{bps1}$ | ZDB-ALT-240703–12 | |
| Strain (*Danio rerio*) | Zebrafish *rpgrip1l*$^{ex4}$ mutant strain | This manuscript, Zfin: *rpgrip1l*$^{bps2}$ | ZDB-ALT-240703–13 | |
| Strain (*Danio rerio*) | Zebrafish *urp2*$^{+/-}$ | *Gaillard et al., 2023*; PMID:36736605 | ZDB-ALT-230207–5 | |
| Strain (*Danio rerio*) | Zebrafish *dnaaf1*$^{tm317b/+}$ | *van Rooijen et al., 2008*; PMID:18385425 | ZDB-FISH-150901–27935 | |
| Strain (*Danio rerio*) | Zebrafish Scospondin-GFP$^{ut24}$ | *Troutwine et al., 2020*; PMID:32386529 | ZDB-FIG-200728–38 | |
| Antibody | anti-Bovine Reissner Fiber (rabbit polyclonal) | *Didier et al., 1995*; PMID:7577440 | Courtesy of Dr. S.Gobron | IF(1:200) |
| Antibody | anti-Human ZO1 (Mouse monoclonal) | Thermofisher | Cat#:33–9100 RRID:AB_2533147 | IF: (1:150) |
| Antibody | anti-human acetylated-tubulin (clone 6-11B-1) (Mouse monoclonal) | Sigma-Aldrich | Cat#:T 6793 RRID:AB_477585 | IF: (1:400) |
| Antibody | anti glutamylated Tubulin (GT335 clone, Mouse monoclonal) | Adipogen | Cat#:AG-20B-0020-C100 | IF: (1:500) |
| Antibody | anti-human Arl13b (Rabbit polyclonal) | Proteintech | Cat#:17711–1-AP | IF: (1:200) |
| Antibody | anti human GFAP (mouse monoclonal) | Sigma | Cat#:G3893 | IF: (1:400) |
| Antibody | Anti GFP (chicken monoclonal) | AVES lab | Cat#:GFP-1020 | IF: (1:200) |
| Antibody | Anti Myc (clone 9B-11, mouse monoclonal) | Cell signaling | Cat#:2276 | IF: (1:200) |
| Antibody | Anti zebrafish LCP1 (rabbit polyclonal) | GeneTex | Cat#:GTX124420 | IF: (1:200) |
| Antibody | anti-human gamma-tubulin (Clone GTU-88, mouse monoclonal) | Sigma | Cat#:T6557 | IF:(1:500) |
| Antibody | Anti-human annexin-2 (rabbit polyclonal) | Proteintech | Cat#:11256–1-AP RRID:AB_2057311 | IF: (1:200) |
| Antibody | Anti-mouse IgG1 Alexa633(goat polyclonal) | Molecular probes | Cat#:A-21126 RRID:AB_2535768 | IF: (1:400) |
| Antibody | Anti-mouse IgG1 Alexa568(goat polyclonal) | Molecular Probes | Cat#:A-21124 RRID:AB_2535766 | IF: (1:400) |
| Antibody | Anti-mouse IgG2a Alexa568(goat polyclonal) | Molecular probes | Cat#:A-21134 RRID:AB_2535773 | IF: (1:400) |
| Antibody | Anti-mouse IgG2a Alexa488(goat polyclonal) | Molecular probes | Cat#:A-21131 RRID:AB_141618 | IF: (1:400) |
| Antibody | Anti-mouse IgG2b Alexa633(goat polyclonal) | Molecular probes | Cat#:A-21146 RRID:AB_2535782 | IF: (1:400) |
| Antibody | Anti-mouse IgG2b Alexa568 (goat polyclonal) | Molecular probes | Cat#:A-21144 RRID:AB_2535780 | IF: (1:400) |
| Antibody | Anti-rabbit IgG Alexa568 (goat polyclonal) | Molecular probes | Cat#:A-11011 RRID:AB_2535780 | IF: (1:400) |

*Continued on next page*

*Continued*

| Reagent type (species) or resource | Designation | Source or reference | Identifiers | Additional information |
|---|---|---|---|---|
| Antibody | Anti-rabbit IgG Alexa488 (goat polyclonal) | Molecular probes | Cat#:A-11008 RRID:AB_143165 | IF: (1:400) |
| Antibody | Anti-Chicken IgY -FITC (donkey polyclonal) | Jackson Immuno-Research | Cat#:703-096-155 RRID:AB_2340357 | IF: (1:200) |
| Recombinant DNA reagent | *foxj1a:: 5xMyc-RPGRIP1L* (Tol2 construct) | This paper | | Built by Gateway method, see Materials and methods, transgenesis section |
| Recombinant DNA reagent | *col2a1a:: 5xMyc-RPGRIP1L* (Tol2 construct) | This paper | | Built by Gateway method, see Materials and methods, transgenesis section |
| Sequence-based reagent | Rpgrip1l_ex25F | This paper | Genotyping primers | AGTGTGCGGTACATCTCCAA |
| Sequence-based reagent | Rpgrip1l-ex4-del F | This paper | Genotyping primers | CCCACACTGCATACGCACTC |
| Sequence-based reagent | Rpgrip1l_ex25_R3 | This paper | Genotyping primers | GTTGTGTCTCTGCCATATATTG |
| Sequence-based reagent | Urotensin2 family neuro-peptides | *Gaillard et al., 2023*; PMID:36736605 | Q-PCR primers | Listed in *Supplementary file 1* of Gaillard et al. |
| Sequence-based reagent | Rpgrip1l_x4_G1 | This paper | Cas9-Guide RNA | GCTTACGGTCCTTCACCAGACGG |
| Sequence-based reagent | Rpgrip1l_x25_G3 | This paper | Cas9-Guide RNA | CCTCAGTTGACAGGTTTCAGCGG |
| Commercial assay or kit | Gateway BP Clonase II Enzyme Mix | Invitrogen | Cat#:11789–020 | |
| Commercial assay or kit | Gateway LR Clonase II Plus Enzyme Mix | Invitrogen | Cat#:12538–120 | |
| Commercial assay or kit | Megascript T7 Transcription Kit | Thermo Fisher Scientific, Waltham, MA | Cat#:AM1334 | |
| Commercial assay or kit | miRNAeasy QIAGEN kit | QIAGEN | Cat#:217004 | |
| Commercial assay or kit | QIAEX II gel extraction kit | QIAGEN, | Cat#:20021 | |
| Commercial assay or kit | KAPA mRNA Hyperprep kit | Roche | Cat#:08098115702 | |
| Chemical compound, drug | NACET | Biolla Chemicals | Cat#:59587-09-6 | |
| Chemical compound, drug | Vectashield | Vector Laboratories | Cat#:H-1000 | |
| Recombinant DNA reagent | Foxj1 promoter-enhancer sequence | *Grimes et al., 2016*; PMID:27284198 | | |
| Recombinant DNA reagent | Col2a1a promter-enhancer | *Dale and Topczewski, 2011*; PMID:21723274 | | |
| Software | LASX | Leica | RRID:SCR_013673 | |
| Software | Amira for Life & Biomedical Sciences | Thermo Fisher Scientific | RRID:SCR_007353 | |
| Software | CRISPOR | *Haeussler et al., 2016* | | http://crispor.tefor.net/ |
| Software | Fiji/ImageJ | *Schindelin et al., 2012* | | https://imagej.net/Fiji/Downloads |
| Software | PRISM | GraphPad | RRID:SCR_002798 | https://www.graphpad.com/ |
| Software | Metascape | | RRID:SCR_016620 | https://metascape.org/gp/index.html#/main/step1 |
| Software | Matlab | The Mathworks, Inc. | RRID:SCR_001622 | https://fr.mathworks.com/products/matlab.html |
| Software | NRecon reconstruction software | Micro Photonics Inc. | | https://www.microphotonics.com/micro-ct-systems/nrecon-reconstruction-software/ |

*Continued on next page*

*Continued*

| Reagent type (species) or resource | Designation | Source or reference | Identifiers | Additional information |
|---|---|---|---|---|
| Software | CTvox | Bruker | | https://www.microphotonics.com/micro-ct-systems/visualization-software/ |
| Software | CT Analysis | Bruker | | https://www.microphotonics.com/micro-ct-systems/ct-analysis/ |
| Other | Refractometer | Roth Sochiel EURL DR 301–95 | | Measure of refraction index of mounting medium used for IF on cleared brains (Materials and methods) |
| Other | Confocal microscope | Leica (SP5) | | Whole mount IF imaging on zebrafish embryos |
| Other | Confocal microscope | Zeiss (LSM 980 Inverted) | | Imaging IF of brain paraffin sections |
| Other | Confocal microscope | Zeiss (LSM 710) | | Imaging IF of brain paraffin sections |
| Other | Macroscope | Zeiss (AXIOZOOM V16) | | Lcp1 positive cells quantification on whole brain |
| Other | Micro-scanner | Bruker (Skyscan 1272) | | See µCT scans in Materials and methods |
| Other | Microtome | Leica (RM2125RT) | | See Paraffin sectioning of brains for IF (Materials and methods) |
| Other | Q-PCR apparatus | Applied Biosystems (Step One plus) | | Q-PCR analysis (Materials and methods) |

## Zebrafish

Wild-type, *rpgrip1l*$^{ex4}$ and *rpgrip1l*$^{\Delta}$ zebrafish embryos and adults were raised, staged and maintained as previously described (*Kimmel et al., 1995*). All our experiments were made in agreement with the european Directive 210/63/EU on the protection of animals used for scientific purposes, and the French application decree 'Décret 2013–118'. The projects of our group have been approved by our local ethical committee 'Comité d'éthique Charles Darwin'. The authorization number is APAFIS #31540–2021051809258003 v4. The fish facility has been approved by the French 'Service for animal protection and health' with approval number A750525. All experiments were performed on *Danio rerio* embryos of mixed AB/TL background. Animals were raised at 28.5 °C under a 14/10 light/dark cycle.

## Rpgrip1l mutant generation and genotyping
### Guide RNA preparation and microinjection
CRISPR target sites were selected for their high predicted specificity and efficiency using the CRISPOR online tool (*Haeussler et al., 2016*). Real efficiency was assessed on zebrafish embryos by T7E1 test. The two most efficient guides Rpgrip1l_x4_G1 and Rpgrip1l_x25_G3 were respectively situated 24 nt from the beginning of exon 4 and 82 nt downstream of exon 25 were kept for further experiments. Their sequence is described in the Key Resource Table. The sgRNA: Cas9 RNP complex was obtained by incubating Cas9 protein (gift of TACGENE, Paris, France;7.5 µM) with sgRNA (10 µM) in 20 mM Hepes-NaOH pH 7.5, 150 mM KCl for 10 min at 28 °C. 1–2 nl of the complex was injected per embryo. To obtain a deletion within *rpgrip1l*, Rpgrip1l_x4_G1 and Rpgrip1l_x25_G3 RNP complexes were mixed half and half.

We kept two F1 fish, one with a 26Kb deletion between exon 4 and 25 of *rpgrip1l*, that we called *rpgrip1l*$^{\Delta}$ and that we submitted to ZFIN under the name *rpgrip1l* $^{bps1}$and a second one with a stop codon within exon 4 called *rpgrip1l* $^{ex4}$, that is submitted to ZFIN under the name *rpgrip1l* $^{bps2}$.

## Screening and genotyping
Injected (F0) fish were screened for germline transmission by crossing with wild type fish and extracting genomic DNA from resulting embryos. For genomic DNA extraction, caudal fin (juveniles/adults) or whole embryo DNA were used. Genomic DNA was isolated with Proteinase K (PK) digestion in 40 of lysis buffer (100 mM Tris-HCl pH 7.5, 1 mM EDTA, 250 mM NaCl, 0.2% SDS, 0.1 µg/µl Proteinase K (PK)) for embryos (300 µl for adult fin) overnight at 37 °C with agitation. PK was inactivated for 10 min at 90 °C and a fivefold dilution was used as template for PCR amplification. Combined genotyping of wild type and mutant alleles (*Figure 1—figure supplement 1A'*) was performed by PCR using three

primers, a common reverse primer for both alleles: Rpgrip1l_ex25_R3, a specific forward primer for the deleted allele: Rpgrip1l-ex4-del F and a specific forward primer for the wild type allele: Rpgrip1l_ex25F at an annealing temperature of 60 °C for 35 cycles. Their sequence is described in the Key Resource Table. The sizes of the amplified fragments are 364 nt for the wild type allele and 544nt for the mutant allele. The deleted allele presents a frameshift at position N77 leading to the generation of a STOP codon after 11 aas, thus leading to a N-terminally truncated protein at position 88/1256 Aas.

For genotyping the *urp2* mutant line, caudal fins from adult fish were extracted in 300 μl of lysis solution and PCR was performed as above to detect exon 5 deletion using the primers described in the Key Resource Table.

## μCT scans

The samples were scanned on a Bruker micro scanner (Skyscan 1272) with a resolution of 8.5 μm, a rotation step of 0.55° and a total rotation of 180°. For the acquisition of adult fish (2.5 cm), a 0.25 mm aluminium filter was used, for a voltage of 50 kV and an intensity of 180 mA, for juvenile fish (0.9–1.2 cm) the filter was omitted, and a voltage of 60 kV was used with an intensity of 166 mA. Each image contained 1008x672 pixels and was based on the average of three images. The 3D reconstruction by backprojection was carried out by the NRecon software and the 2D image overlays were then cleaned by the CT Analyser software. The Dataviewer software allowed all fish skeletons to be oriented in the same way taking the otoliths as landmarks. The CTvox software then allowed 3D visualizing of the samples. Morphometric analysis was performed with the CT Analyser software.

## Scanning electron microscopy

Three-month fish (three controls, five mutants) were euthanized using lethal concentration of MS222 (0.028 mg/mL). The brains were quickly dissected in 1.22 X PBS (pH 7.4), 0.1 M sodium cacodylate and fixed overnight with 2% glutaraldehyde in 0.61 X PBS (pH 7.4), 0.1 M sodium cacodylate at 4 °C. They were sectioned along the dorsal midline with a razor blade to expose their ventricular surfaces. Both halves were washed four times in 1.22 X PBS and post-fixed for 15 min in 1.22 X PBS containing 1% OsO4. Fixed samples were washed four times in ultrapure water, dehydrated with a graded series of ethanol and critical point dried (CPD 300, Leica) at 79 bar and 38 °C with liquid CO2 as the transition fluid and then depressurized slowly (0.025 bar/s). They were then mounted on aluminum mounts with conductive silver cement. Sample surfaces were coated with a 5 nm platinum layer using a sputtering device (ACE 600, Leica). Samples were observed under high vacuum conditions using a Field Emission Scanning Electron Microscope (Gemini 500, Zeiss) operating at 5 kV, with a 20 μm objective aperture diameter and a working distance around 3 mm. Secondary electrons were collected with an in-lens detector. Scan speed and line compensation integrations were adjusted during observation.

## Histological sections of juvenile and immunofluorescence on sections

Juvenile and adult zebrafish were euthanized using lethal concentration of MS222 (0.028 mg/mL). Pictures and size measurements were systematically taken before fixation. Fish were fixed in Zamboni fixative [35 ml PFA 4 %, 7.5 ml saturated picric acid (1.2 %), 7.5 ml 0.2 M Phosphate Buffer (PB)] [55] overnight at 4 °C under agitation. Fish were washed with Ethanol 70% and processed for dehydration by successive 1 hr incubation in Ethanol (3x70% and 2x100%) at room temperature under agitation, then for paraffin inclusion. 14 μm sagittal paraffin sections were obtained using a Leica RM2125RT microtome. Sections were deparaffinized and antigen retrieval was performed by incubation for 7 min in boiling citrate buffer (10 mM, pH 6). Immunofluorescence staining was performed as described previously (*Andreu-Cervera et al., 2019*). The following primary antibodies were used: anti-RF, anti-Acetylated Tubulin, anti-Glutamylated Tubulin, anti-Arl13b, anti-LCP1, anti-AnnexinA2. Corresponding primary and secondary antibodies are described in the Key Resource Table.

## Immunofluorescence on whole embryos

Embryos from 24 to 40 hpf were fixed 4 hr to overnight in 4% paraformaldehyde (PFA) at 4 °C. For Reissner fiber staining, larvae at 48 and 72 hpf were fixed 2 hr in 4% PFA and 3% sucrose at 4 °C, skin from the rostral trunk and yolk were removed. Samples from 24 to 40 hpf embryos were blocked overnight in a solution containing 0.5% Triton, 1% DMSO, 10% normal goat serum and 2 mg/mL BSA. For older samples (48–72 hpf) triton concentration was increased to 0.7%. Primary antibodies

were incubated one to two nights at 4 °C in a buffer containing 0.5% Triton, 1% DMSO, 1% NGS and 1 mg/mL BSA. All secondary antibodies were from Molecular Probes, used at 1:400 in blocking buffer, and incubated a minimum of 2.5 hr at room temperature. The primary antibodies chosen for in toto immuno-labeling against Reissner fiber, Acetylated-tubulin, Myc, Arl13b and Gamma-tubulin as well as the corresponding secondary antibodies are referenced in the Key Resources Table. Whole mount zebrafish embryos (dorsal or lateral mounting in Vectashield Mounting Medium) were imaged on a Leica SP5 confocal microscope and Zeiss LSM910 confocal microscope, both equipped with a 63 X immersion objective. Images were then processed using Fiji (*Schindelin et al., 2012*).

## Whole-mount brain clearing

Brains were dissected from 4 to 5 wpf size-matched (0.9–1.2 mm length) zebrafish after in toto fixation with formaldehyde. Whole–mount tissue clearing was performed following the zPACT protocol (*Affaticati et al., 2017*). In brief, brains were infused for 2 days in hydrogel monomer solution (4% acrylamide, 0.25% VA- 044, 1% formaldehyde and 5% DMSO in 1 X PBS) at 4 °C. Polymerization was carried out for 2hr 30 min at 37 °C in a desiccation chamber filled with pure nitrogen. Brains were transferred into histology cassettes and incubated in clearing solution (8% SDS and 200 mM boric acid in dH2O) at 37 °C with gentle agitation for 8 days. Cleared brains were washed in 1 X PBS with 0.1% Tween-20 (PBT) for 3 days at room temperature and kept in 0.5% formaldehyde, 0.05% sodium azide in PBT at 4 °C until further processing. Brains were subsequently placed for 1 hr in depigmentation pre-incubation solution (0.5 X SSC, 0.1% Tween-20 in dH2O) at room temperature. The solution was replaced by depigmentation solution (0.5 X SSC, Triton X-100 0.5%, formamide 0.05% and $H_2O_2$ 0.03% in dH2O) for 45 min at room temperature. Depigmented brains were washed for 4 hr in PBT and post-fixed (2% formaldehyde and 2% DMSO in PBT) overnight at 4 °C.

## Cleared brain immunostaining of whole adult brains

Whole-mount immunolabeling of cleared brains was performed as described in the zPACT protocol with slight modifications. Briefly, brains were washed in PBT for 1 day at room temperature and blocked for 10 hr in 10% normal goat serum, 10% DMSO, 5% PBS-glycine 1 M and 0.5% Triton X-100 in PBT at room temperature. Brains were washed again in PBT for 1 hr prior to incubation with anti- ZO-1 antibody (ZO1-1A12, Thermofisher, 1:150) in staining solution (2% normal goat serum, 10% DMSO, 0.1% Tween-20, 0.1% Triton X-100 and 0.05% sodium azide in PBT) for 12 days at room temperature under gentle agitation. Primary antibody was renewed once after 6 days of incubation. Samples were washed three times in PBT and thereafter incubated with Alexa Fluor 488-conjugated secondary antibody (A11001, Invitrogen, 1:200) for 10 days in the staining solution at room temperature under gentle agitation. Secondary antibody was renewed once after 5 days of incubation. Samples were washed three times in PBT prior to a counterstaining with DiIC18 (D282, Invitrogen, 1 μM) in the staining solution for 3 days at room temperature with gentle agitation. Samples were washed in PBT for 3 hr before mounting procedure.

## Mounting and confocal imaging

Samples were placed in 50% fructose-based high-refractive index solution fbHRI, see *Affaticati et al., 2017* /50% PBT for 1 hr, then in 100% fbHRI. Brains were mounted in agarose-coated (1% in standard embryo medium) 60 mm Petri dish with custom imprinted niches to help orientation. Niche-fitted brains were embedded in 1% phytagel and the Petri dish filled with 100% fbHRI. fbHRI was changed three times before imaging until its refractive index matched 1.457. Images were acquired with a Leica TCS SP8 laser scanning confocal microscope equipped with a Leica HC FLUOTAR L 25 x/1.00 IMM motCorr objective. Brains were scanned at a resolution of 1.74x1.74 × 1.74 μm (xyz) and tiled into 45–70 individual image stacks, depending on brain dimensions, subsequently stitched, using LAS X software.

## Volumetric analysis of the posterior ventricles

The volumes of the posterior ventricles were segmented, reconstructed and analyzed using Amira for Life & Biomedical Sciences software (Thermo Fisher Scientific). In essence, the ventricles volumes were manually segmented in Amira's segmentation editor and subsequently refined by local thresholding and simplification of the corresponding surfaces. Volumes, which were open to the environment were

artificially closed with minimal surfaces by connecting the distal-most points of their surface to the contralaterally corresponding points using straight edges. Due to the biologic variability of the sample population, the overall size of the specimens needed to be normalized to keep the measured volumes comparable. For this spatial normalization, one of the specimens was randomly selected from the wildtype population as template (1664, grey) and the 'Registration' module in Amira was used to compute region-specific rigid registrations for the other specimens, allowing for isotropic scaling only. For excluding the influence of the ventricular volumes, the registration was computed on the basis of the independent reference stain (DiIC18). Region specific volume differences between the mutant and wildtype population were evaluated on seven subvolumes of the posterior ventricles.

## RNA extraction for transcriptome analysis and quantitative RT-PCR

Juvenile and adult zebrafish were euthanized using lethal concentration of MS222 (0.28 mg/mL). Their length was measured and their fin cut-off for genotyping. For transcriptomic analysis, brain and dorsal trunk from 1 month juvenile (0.9–1.0 cm) zebrafish were dissected in cold PBS with forceps and lysed in QIAzol (QIAGEN) after homogenization with plunger pistons and 1 ml syringes. Samples were either stored in QIAzol at –80 °C or immediately processed. Extracts containing RNA were loaded onto QIAGEN-mini columns, DNAse digested and purified in the miRNAeasy QIAGEN kit (Cat 217004) protocol. Samples were stored at –80 °C until use. RNA concentration and size profile were obtained on the Tapestation of ICM platform. All preparations had a RIN above 9.2. For Q-PCR from juveniles, whole fish minus internal organs were lysed in Trizol (Life technologies) using the Manufacturer protocol.

## Quantitative PCR from individual juvenile or adult fish

The cDNA from the isolated RNAs was obtained following GoScript Reverse Transcription System protocol (Promega), using 3–4 µg of total RNA for each sample. The 20 µL RT-reaction was diluted fourfold and 4 µl was used for each amplification performed in duplicates. The qPCR primers for *urp2*, *urp1*, *urp*, *uts2a* are those described in Table S1 from *Gaillard et al., 2023*, as well as that of the reference gene *lsm12b*. The Q-PCR were performed using the StepOne real-time PCR system and following its standard amplification protocol (Thermo Fisher Scientific). Relative gene expression levels were quantified using the comparative CT method ($2^{-\Delta\Delta CT}$ method or $2^{-\Delta CT}$) on the basis of CT values for target genes and *lsm12b* as internal control.

## RNA-sequencing and analysis

RNA-seq libraries were constructed by the ICM Platform (PARIS) from 100 ng of total RNA using the KAPA mRNA Hyperprep kit (Roche) that allows to obtain stranded polyA libraries, and their quality controlled on Agilent Tape station. The sequencing was performed on a NovaSeq 6000 Illumina for both strands. Sequences were aligned against the GRCz11 version of zebrafish genome using *RNA Star function* in 'Galaxy' environment. The 12 Brain libraries reached between 22–26 Million of assigned reads, and the trunk libraries between 26–35 Millions of assigned reads. Expression level for each gene was determined using *Feature counts function* from RNA Star BAM files. *DESeq2 function* was used to calculate the log2 Fold change and adjusted P values from the Wald test for each gene comparing the seven mutants raw count collection to the five controls raw counts collection. Volcano plots were generated using the Log2 fold change and P adj. value of each Gene ID from the DESeq2 table. A threshold of 5.10 E-2 was chosen for the Padj. values and a log2 fold change >to + 0.75 or<to –0.75 to select significantly up or down-regulated regulated genes. Both gene lists were used as input to search for enriched GO terms and KEGGs pathways using Metascape software (*Zhou et al., 2019*) as well as to compare up-regulated genes in the dorsal trunk versus the brain using Galaxy.

## Quantitative proteomics

### Sample preparation

Adult zebrafish were euthanized using lethal concentration of MS222 (0.28 mg/mL). For proteomic analysis, brain from 3 months zebrafish were dissected in cold PBS with forceps and immediately flash frozen in liquid nitrogen and stored at –80 °C. Zebrafish brain samples were lysed in RIPA buffer (Sigma) with 1/100 antiprotease and sonicated for 5 min. After a 660 nm protein assay (Thermo), samples were digested using a Single Pot Solid Phase enhanced Sample Preparation (SP3) protocol

(*Hughes et al., 2014*). In brief, 40 µg of each protein extract was reduced with 12 mM dithiothreitol (DTT) and alkylated using 40 mM iodoacetamic acid (IAM). A mixture of hydrophilic and hydrophobic magnetic beads was used to clean-up the proteins at a ratio of 20:1 beads:proteins (Sera-Mag Speed beads, Thermo Fisher Scientific). After addition of ACN to a final concentration of 50%, the beads were allowed to bind to the proteins for 18 min. Protein-bead mixtures were washed twice with 80% EtOH and once with 100% ACN. The protein-bead complexes were digested with a mixture of trypsin:LysC (Promega) at a 1:20 ratio overnight at 37 °C. Extracted peptides were cleaned-up using automated C18 solid phase extraction on the Bravo AssayMAP platform (Agilent Technologies).

## NanoLC-MS-MS analysis

The peptide extracts were analysed on a nanoLC-TimsTof Pro coupling (Bruker Daltonics). The peptides (200 ng) were separated on an IonOpticks column (25 cm X 75 µm 1.6 µm C18 resin) using a gradient of 2–37% B (2%ACN, 0.1%FA) in 100 min at a flow rate of 0.3 µl/min. The dual TIMS had a ramp time and accumulation time of 166ms resulting in a total cycle time of 1.89 s. Data was acquired in Data Dependent Acquisition-Parallel Accumulation Serial Fragmentation (DDA-PASEF) mode with 10 PASEF scans in a mass range from 100 m/z to 1700 m/z. The ion mobility scan range was fixed from 0.7 to 1.25 Vs/cm$^2$. All samples were injected using a randomized injection sequence. To minimize carry-over, one solvent blank injection was performed after each sample.

## Data interpretation

Raw files were converted to.mgf peaklists using DataAnalysis (version 5.3, Bruker Daltonics) and were submitted to Mascot database searches (version 2.5.1, MatrixScience, London, UK) against a *Danio rerio* protein sequence database downloaded from UniProtKB-SwissProt (2022_05_18, 61 732 entries, taxonomy ID: 7955), to which common contaminants and decoy sequences were added. Spectra were searched with a mass tolerance of 15 ppm in MS mode and 0.05 Da in MS/MS mode. One trypsin missed cleavage was tolerated. Carbamidomethylation of cysteine residues was set as a fixed modification. Oxidation of methionine residues and acetylation of proteins' n-termini were set as variable modifications. Identification results were imported into the Proline software (version 2.1.2, http://www.profiproteomics.fr/proline/) (*Bouyssié et al., 2020*) for validation. Peptide Spectrum Matches (PSM) with pretty ranks equal to one were retained. False Discovery Rate (FDR) was then optimized to be below 1% at PSM level using Mascot Adjusted E-value and below 1% at protein level using Mascot Mudpit score. For label free quantification, peptide abundances were extracted without cross assignment between the samples. Protein abundances were computed using the sum of the unique peptide abundances normalized at the peptide level using the median.

To be considered, proteins must be identified in at least four out of the five replicates in at least one condition. Imputation of the missing values and differential data analysis were performed using the open-source ProStaR software (*Wieczorek et al., 2017*). Imputation of missing values was done using the approximation of the lower limit of quantification by the 2.5% lower quantile of each replicate intensity distribution ('det quantile'). A Limma moderated *t*-test was applied on the dataset to perform differential analysis. The adaptive Benjamini-Hochberg procedure was applied to adjust the p-values and FDR. The mass spectrometry proteomics data have been deposited to the ProteomeXchange Consortium via the PRIDE partner repository (*Perez-Riverol et al., 2022*), with the dataset identifier PXD042283.

## Embryos drug treatment

Twenty-seven hpf embryos from *dnaaf1*[tm317b/+] incross were dechorionated manually. N-acetyl cysteine ethyl ester (NACET) (BIOLLA Chemicals #59587-09-6) was prepared extemporaneously at the final concentration of 3 mM in pure water (pH: 7.2; conductivity 650 µS). Embryos were treated between 27 hpf to 60 hpf and embryos were anesthetized before being imaged laterally.

## Juvenile drug treatment and preparation of treated samples

Two hundred larvae were housed off-system in 1.8 L tanks with 20 fish per tank, being fed twice a day throughout the experiment. A total of 100 were treated with NACET which was prepared extemporaneously at 1.5 mM (286.5 mg/L) in fish water (pH 7.2, Conductivity 650 µS) which was changed once per day. Fish were treated from 30 dpf to 84 dpf and monitored for curvature onset once per week.

At 85 dpf, fish were euthanized, imaged to measure the curvature index. Half of the fish were fixed for histology analysis in zamboni fixative (35 ml PFA 4 %, 15 ml saturated picric acid [1.2 %]) overnight at 4 °C under agitation, while the other half was processed for RT-qPCR analysis. To prepare dorsal trunk RNA, internal organs were removed in cold PBS 1 X, and the remaining tissue was cut in small pieces before lysis in QIAzol (QIAGEN). Homogenization was achieved using plunger pistons and passages through 1 ml syringes.

## Transgenesis

The *Tol2 –5.2foxj1a::5xmyc-RPGRIP1L* plasmid was produced using the Gateway system by combining four plasmids. P5E-foxj1a enhancer was kindly donated by B. Ciruna and described in *Grimes et al., 2016*. 5xmyc-RPGRIP1L cDNA was amplified from the plasmid pCS2-5xmyc-RPGRIP1L (*Mahuzier et al., 2012*) using CloneAmp HiFi PCR Premix (Takara # 639298) using the primers: MeRPGRIP1L-F and MeRPGRIP1L-R described in Key Resource Table. The PCR product was gel purified using QIAEX II gel extraction kit (QIAGEN, #20021). BP recombination (Invitrogren 'Gateway BP Clonase II Enzyme Mix' #11789–020) was performed into pDONR221 #218 to generate pMe-5xMyc-RPGRIP1L. The 3' entry polyadenylation signal plasmid was the one described in [61]. The three plasmids were shuttled into the backbone containing the cmcl2:GFP selection cassette (*Kwan et al., 2007*) using the LR recombination (Invitrogen 'Gateway LR Clonase II Plus Enzyme Mix' #12538–120). *rpgrip1l*+/-X AB embryos outcrosses were injected at the one cell stage with 1 nl of a mix containing 20 ng/µl plasmid and 25 ng/µl Tol2 transposase RNA and screened at 48 hpf for GFP expression in the heart. Some F0 founders produced F1 zebrafish carrying one copy of the transgene that were further checked for RPGRIP1L specific expression in *foxj1a* territory, using Myc labelling at the base of FP cilia in 2.5 dpf embryos.

The Tol2-1.7kb col2a1a:5xMyc-RPGRIP1L plasmid was produced using the Gateway System by combining four plasmids. Col2a1a enhancer was amplified from the plasmid –1.7kbcol2a1a:EGFP-CAAX (*Dale and Topczewski, 2011*) using CloneAmp HiFi premix (Takara #639298) with the primers: p5E-Col2a1a-F and p5E-Col2a1a-R. The PCR product was gel purified and BP recombination was performed into pDONR_P4-P1R-#219. The middle entry plasmid and the 3' plasmid were the same as those used to generate the Tol-5.2foxj1a: 5xmyc-RPGRIP1L plasmid. F1 transgenic were first selected for GFP expression in the heart and then on Myc labelling within notochordal cells at 2.5 dpf within their embryonic progeny.

## Skeletal preparations, Alizarin staining and imaging

Animals were euthanized using a lethal concentration of MS222 (0.28 mg/mL), skin was removed and fixation was performed with 4% paraformaldehyde overnight at 4 °C. After evisceration, samples were incubated in borax 5%, rinsed several times in 1% KOH and incubated in solution composed with Alizarin 0.01% (Sigma, A5533) and KOH 1% during 2 days. Fish were rinsed several times in KOH 1% and incubated in trypsin 1% and borax 2% for 2 days until cleared. Samples were rinsed several times in distilled water and transferred in glycerol 80% (in KOH 1%) using progressive dilutions. Samples were kept in glycerol 80% until imaging.

## Cobb angles measurements

To quantitatively evaluate the severity of spine curvature in *rpgrip1l*+/-; *urp2*+/- +/-, we used the zebrafish skeletal preparations stained with Alizarin. We drew parallel lines to the top and bottom most displaced vertebrae on lateral and dorsal views. The Cobb angle was then measured as the angle of intersection between lines drawn perpendicular to the original 2 lines. This was conducted using FIJI software. For each fish, the total Cobb angle was calculated by summing all Cobb angles.

### Curvature index measurements

We implemented a novel procedure to quantify curvatures on both axes by drawing a line along the body of the fish as shown in *Figure 4—figure supplement 1I* and curvature was calculated in MATLAB (code available on demand). We decomposed the line in a series of equidistant points and we measured the curvature at each of these points using the LineCurvature2D function. We then added the absolute values of these measures, which are expressed as an angle (in radian) per unit of length. The higher the sum is for a given line, the more this line is curvated. We verified the accuracy of this

metric by comparing it with the visual assessment of curvatures by three independent observers. For *rpgrip1l*[+/-]; *urp2*[+/-] incross and NACET experiment, a line following the body deformation was drawn from the mouth to the base of the tail following the midline of the fish in lateral position and another one along the dorsal axis. Curvature index of both curves were summed for each fish. Curvature analysis was performed blinded to fish genotype. *rpgri1l*[+/+] or *rpgrip1l*[+/-] siblings were used as controls.

## Data acquisition for body-curvature analysis at embryonic stage

Zebrafish embryos were anesthetized and imaged laterally with the head pointing to the left. The angle between the line connecting the center of the eye to the center of the yolk and the line connecting the center of the yolk to the tip of the tail was measured to evaluate the body curvature of the embryos. For quantitative analysis, the angles of each embryo were put in the same concentric circles represented with a Roseplot, with 0° pointing to the right and 90° pointing to the top. Each triangle represents a 30° quadrant, and its height indicates the number of embryos within the same quadrant.

## Quantification, statistical analysis, and figure preparation

For all experiments the number of samples analyzed is indicated in the text and/or visible in the figures. Statistical analysis was performed using the Prism software. ****: p-value <0.0001; ***: p-value <0.001; **: p-value <0.01; *: p-value <0.05. Graph were made using Prism and Matlab and figures were assembled using Photoshop software.

## Acknowledgements

We are grateful to the IBPS aquatic animal, imaging and bioinformatics facilities and to the ICM sequencing facility for their technical assistance. We thank Michaël Trichet from the IBPS electron microscopy facility for participating in the MEB experiments, the TACGENE facility for providing the Cas9 protein; the TEFOR Paris-Saclay facility for the brain clearing experiment; Thierry Jaffredo and Pierre Charbord for their precious help in transcriptome analysis; Martin Catala for helpful discussions on patients with Chiari malformation presenting syringomyelia, Nicolas Baylé for initial characterization of the *rpgrip1l* mutant; Claudia Hoffman for providing *Cep290* fixed samples; Brian Ciruna for sharing the *foxj1a* expressing constructs and detailed NACET treatment protocol; Ryan Gray for sharing the *scospondin-GFP*[ut24] zebrafish line. This work was supported by funding to SSM from the Fondation pour la Recherche Médicale (Equipe FRM EQU201903007943) and the Fondation Yves Cotrel.

## Additional information

### Funding

| Funder | Grant reference number | Author |
| --- | --- | --- |
| Fondation pour la Recherche Médicale | Equipe FRM EQU201903007943 | Sylvie Schneider-Maunoury |
| Fondation Yves Cotrel | | Sylvie Schneider-Maunoury |

The funders had no role in study design, data collection and interpretation, or the decision to submit the work for publication.

### Author contributions

Morgane Djebar, Conceptualization, Data curation, Formal analysis, Validation, Investigation, Visualization, Methodology, Writing – original draft; Isabelle Anselme, Validation, Investigation; Guillaume Pezeron, Resources, Formal analysis, Writing – review and editing; Pierre-Luc Bardet, Software, Formal analysis, Methodology, Writing – review and editing; Yasmine Cantaut-Belarif, Resources, Writing – review and editing; Alexis Eschstruth, Resources, Validation, Methodology; Diego López-Santos, Investigation; Hélène Le Ribeuz, Formal analysis, Investigation; Arnim Jenett, Data curation, Formal analysis, Investigation, Visualization, Methodology; Hanane Khoury, Resources, Formal analysis; Joelle Veziers, Data curation, Formal analysis, Methodology; Caroline Parmentier, Resources,

Investigation, Methodology; Aurélie Hirschler, Data curation, Formal analysis, Validation, Investigation, Methodology; Christine Carapito, Resources, Software, Formal analysis, Validation, Investigation, Methodology; Ruxandra Bachmann-Gagescu, Resources, Investigation, Writing – review and editing; Sylvie Schneider-Maunoury, Conceptualization, Funding acquisition, Methodology, Writing – original draft, Project administration, Writing – review and editing; Christine Vesque, Conceptualization, Resources, Data curation, Formal analysis, Supervision, Funding acquisition, Validation, Investigation, Visualization, Methodology, Writing – original draft, Project administration, Writing – review and editing

### Author ORCIDs

Christine Carapito https://orcid.org/0000-0002-0079-319X
Sylvie Schneider-Maunoury http://orcid.org/0000-0002-0797-4735
Christine Vesque https://orcid.org/0000-0001-7983-4953

### Ethics

All our zebrafish experiments were made in agreement with the european Directive 210/63/EU on the protection of animals used for scientific purposes, and the French application decree Décret 2013-118'. The projects of our group have been approved by our local ethical committee 'Comité d'éthique Charles Darwin'. The authorization number is APAFIS #31540-2021051809258003 v4. The fish facility has been approved by the French 'Service for animal protection and health' with approval number A750525.

Reviewer #1 (Public Review): https://doi.org/10.7554/eLife.96831.3.sa1
Reviewer #2 (Public Review): https://doi.org/10.7554/eLife.96831.3.sa2
Author response https://doi.org/10.7554/eLife.96831.3.sa3

## Additional files

### Supplementary files

• Supplementary file 1. Excel table providing the list of differentially expressed genes within the dorsal trunk of rpgrip1l-/- fish versus controls at 5 wpf.

• Supplementary file 2. Excel table providing the list of differentially expressed genes within the brain of *rpgrip1l-/-* fish versus controls at 5 wpf.

• Supplementary file 3. List of the Foxj1 direct (first tab) and indirect (second tab) target genes upregulated in the *rpgrip1l-/-* transcriptomes compared to controls.

• Supplementary file 4. Excel file providing the list of deregulated proteins in the *rpgrip1l-/-* proteome compared to controls.

• Supplementary file 5. Graphical abstract.

• MDAR checklist

### Data availability

All transcriptomic data generated or analysed during this study are included in the manuscript as *Supplementary file 1* for 5 weeks Dorsal trunk samples and as *Supplementary file 2* for 5 weeks Brain samples. The mass spectrometry proteomics data have been deposited to the ProteomeXchange Consortium via the PRIDE partner repository (*Perez-Riverol et al., 2022*), with the dataset identifier PXD042283.

The following dataset was generated:

| Author(s) | Year | Dataset title | Dataset URL | Database and Identifier |
|---|---|---|---|---|
| Hirschler A, Christine C | 2024 | Astrogliosis And Neuroinflammation Underlie Scoliosis Upon Cilia Dysfunction | http://www.ebi.ac.uk/pride/archive/projects/PXD042283 | PRIDE, PXD042283 |

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
