## [Editor Report · eLife assessment]

This **valuable** study analyzes the role of rpgrip1l encoding a ciliary transition zone component in the development of neuroinflammation and scoliotic phenotypes in zebrafish. Through proteomic and experimental validation in vivo, the authors demonstrated increased Annexin A2 expression and astrogliosis in the brains of scoliosis fish. Anti-inflammatory drug treatment restored normal spine development in these mutant fish, thus providing additional **convincing** evidence for the role of neuroinflammation in the development of scoliosis in zebrafish.

---

## [Referee Report · Reviewer #1 (Public Review)]

Summary:

In this study, Djebar et al. perform a comprehensive analysis of mutant phenotypes associated with the onset and progression of scoliosis in zebrafish ciliary transition zone mutants rpgrip1l and cep290. They determine that rpgrip1l is required in foxj1a-expressing cells for normal spine development, and that scoliosis is associated with brain ventricle dilations, loss of Reissner fiber polymerization, and the loss of 'tufts' of multi-cilia surrounding the subcommissural organ (the source of Reissner substance). Informed by transcriptomic and proteomic analyses, they identify a neuroinflammatory response in rpgrip1l and cep290 mutants that is associated with astrogliosis and CNS macrophage/microglia recruitment. Furthermore, anti-inflammatory drug treatment reduced scoliosis penetrance and severity in rpgrip1l mutants. Based on their data, the authors propose a feed-forward loop between astrogliosis, induced by perturbed ventricular homeostasis, and immune cells recruitment as a novel pathogenic mechanism of scoliosis in zebrafish ciliary transition zone mutants.

Strengths:

- Comprehensive characterization of the causes of scoliosis in ciliary transition zone mutants rpgrip1l and cep290

- Comparison of rpgrip1l mutants pre- and post-scoliosis onset allowed authors to identify specific phenotypes as being correlated with spine curvature, including brain ventricle dilations, loss of Reissner fiber, and loss of cilia in proximity to the subcommissural organ

- Elegant genetic demonstration that increased urotensin peptide levels do not account for spinal curvature in rpgrip1l mutants

- The identification of astrogliosis and Annexin over-expression in glial cells surrounding diencephalic and rhombencephalic ventricles as being correlated with scoliosis onset and severe curve progression is a very interesting finding, which may ultimately inform pathogenic mechanisms driving spine curvature

Weaknesses:

- The fact that cilia loss/dysfunction and Reissner fiber defects cause scoliosis in zebrafish is already well established in the literature, as is the requirement for cilia in foxj1a-expressing cells

- Neuroinflammation has already been identified as the underlying pathogenic mechanism in at least 2 previously published scoliosis models (zebrafish ptk7a and sspo mutants)

- Anti-inflammatory drugs like aspirin, NAC and NACET have also previously been demonstrated to suppress scoliosis onset and severe curve progression in these models

Therefore, although similar observations in rpgrip1l and cep290 mutants (as reported here) add to a growing body of literature that supports a common biological mechanism underlying spine curvature in zebrafish, novelty of reported findings is diminished.

- Although authors demonstrate that astrogliosis and/or macrophage or microglia cell recruitment are correlated with scoliosis, they do not formally demonstrate that these events are sufficient to drive spine curvature. Thus, the functional consequences of astrogliosis and microglia infiltration remain uncertain.

- Authors do not investigate the effect of anti-inflammatory treatments on other phenotypes they have correlated with spinal curve onset (like ventricle dilation, Reissner fiber loss, and multi-cilia loss around the subcommissural organ). This would help to identify causal events in scoliosis.

---

## [Referee Report · Reviewer #2 (Public Review)]

Summary:

The manuscript by Djebar et al investigated the role and the underlying mechanism of the ciliary transition zone protein Rpgrip1l in zebrafish spinal alignment. They showed that rpgrip1l mutant zebrafish develop a nearly full penetrance of body curvature at juvenile stages. The mutant fish have cilia defects associated with ventricular dilations and loss of the Reissner fibers. Scoliosis onset and progression are also strongly associated with astrogliosis and neuroinflammation, and anti-inflammatory drug treatment prevents scoliosis in mutant zebrafish, suggesting a novel pathogenic mechanism for human idiopathic scoliosis. This study is quite comprehensive with high quality data, and the manuscript is well written, providing important information on how the ciliary transition zone protein functions in maintaining the zebrafish body axis straightness.

Strengths:

Very clear and comprehensive analysis of the mutant zebrafish.

---

## [Author Response]

The following is the authors’ response to the original reviews.

(1) Please provide more background about Rpgrip1l in the introduction, particularly the past studies of mammalian homolog of Rpgrip11, if any? Is there any human disease associated with Rpgrip1l? Do these patients have scoliosis phenotype?

• We have added more background on the human ciliopathies caused by *RPGRIP1L* mutations and on their occasional association with early onset scoliosis (lines 45-54 page 2 in the introduction, see cited references).

(2) The allele is a large deficiency of most of the coding region of rpgrip1l, can you give details in the Supplementary data of how you show this by genotyping? It would be good to explain that this mutation is most likely behaving as a null, if you have RNAseq data that supports this please note that. Otherwise, it may be incorrect to assume it is a null allele as your shorthand nomenclature states. If you do not have stronger evidence that the deficiency allele is behaving as a null allele, then please think about using an allele nomenclature as outlined at ZFIN:

• We now describe in the results section (Lines 72-76, page 3) the extent of the deletion of *rpgrip1l ∆/∆* (22 exons out of 26) that creates an early stop at position 88 of 1256 aas. We have submitted to ZFIN our two novel mutant lines: *rpgrip1l∆* is recorded as *rpgrip1l bps1* and *rpgrip1l ex4* as *rpgrip1l bps2* , and we provide this information in the text. Transcriptomics data confirmed this allele is behaving as a null as the most down-regulated transcript found in the brain of *rpgrip1l ∆/∆* is *rpgrip1l* transcript itself, (volcano plot in Fig 5A, described in the results, Line 270-71, page 9).

• We also have provided in Supplementary Figure 1 A’ a picture of a typical genotyping gel for the *rpgrip1l∆* allele. Sequences of both CRISPR guide RNAs and genotyping primers are provided in the Math & Meth section.

(3) Throughout the manuscript, the authors refer to zebrafish mutant phenotypes as "juvenile scoliosis". However, scoliosis may not appear until 11 weeks post-fertilization in some animals. After 6-8 weeks of age, it would be more appropriate to describe the phenotype as "late-onset or adult scoliosis" to differentiate between other reported scoliosis mutants (such as hypomorphic or dominant negative alleles of scospondin) that start body curvatures at 3-5 dpf .

• We think we can really qualify *rpgrip1l-*/- scoliosis as being a “juvenile scoliosis” as shown by the time course displayed in Fig 1B: *rpgrip1l-*/- scoliosis develops asynchronously between 4 weeks and 9 weeks (from 0.8 cm/1 cm to 1.6 cm, corresponding to juvenile stages according to Parichy et al, 2009 PMID: 19891001), after which it reaches a plateau. Half of the mutants are already scoliotic by 5 weeks and no scoliosis develops at adult stage, ie from 10 weeks on. We have acknowledged the late onset scoliosis in page 3 line 93.

(4) A more careful demonstration of the individual vertebrae, using magnified high-resolution pictures in Figures 1D-G, should be made to more clearly show no obvious vertebral malformations are present.

• We now provide a movie in Sup Data that presents 3D views of controls and mutant spines, which show the intervertebral spaces as well as vertebral shape and size. With these images we could exclude vertebral fusion and the presence of dysmorphic vertebrae.

(5) On page 5: the authors comment on transgenic expression of RPGRIP1L in foxj1a-lineages as "rescuing" scoliosis. This terminology is confusing, as rescuing a condition could be interpreted as inducing it where it was once absent. "Suppressing" scoliosis may be a more appropriate term.

• We agree with the reviewers, the “rescue” term is confusing, we changed it for “suppress” in the title of the paragraph (line 95 page 3) and within the text (line 115 page 3).

(6) On page 5, lines 155-156: the authors state that "Indeed, no tissue-specific rescue has been performed yet in zebrafish ciliary gene mutants". This is misleading, as ptk7a and katnb1 mutations both disrupt cilia, and transgenic reintroduction of both *ptk7a* and *katnb1* in *foxj1a*- expressing lineages has previously been shown to suppress cilia defects as well as scoliosis in these models. The statement should be removed for accuracy.

• We agree that we were not precise enough in our sentence: when we mentioned “ciliary gene” mutants, we were referring to genes whose products are enriched within cilia and directly affecting ciliogenesis, cilia content and maintenance such as TZ or BBS genes, without encompassing genes like *ptk7* and *katnb1* whose products perform multiple functions on top of cilia maintenance such as Wnt signalling and remodelling of the whole microtubule network respectively. We have therefore modified our sentence by adding zebrafish ciliary “TZ and BBS” genes (line 104, page 4).

(7) Figure 2: panels A-B: In the text (line 196) you state that cilia length was increased and that Arl13b content was severely reduced. However, Panel B shows no significant length difference between scoliotic mutants and controls. This statement and graph should be corrected for accuracy. Also, the Arl13b staining is difficult to see in panel A - can channels be split, and/or quantified?

• We have now split the Arl13b and glutamylated tubulin channels (Fig 2 A-C”). We think that the reduction of Arl13b staining intensity is now obvious in both straight and scoliotic mutants (Compare 2A” with 2B” and 2C”). We were not able to quantify Arl13b staining using ciliary masks from glutamylated tubulin staining since both staining only partially overlap along the length of the cilium, Arl13b being more distal than glutamylated tubulin (Fig 2A’).

• Ciliary length was significantly increased (from 3.4 to 5.3 µ) in straight *rpgrip1l-*/-, while the average mean values for scoliotic *rpgrip1l-*/- were heterogenous (mean 4.1µ) and therefore not significantly different when compared to controls. This heterogeneity stems from the combined presence of both shorter and longer cilia in scoliotic fish, a finding we interpreted by the potential breakage over time of extra-long and thin cilia observed in scoliotic fish (as in Sup figure 1 H’’’, Sup Fig 2M’ and 2O’).

• We changed the text to be more accurate: we now state that cilia length increased in straight mutants, and became more heterogenous than controls in scoliotic mutants (line 143-144, page 5).

(8) Figure 3: Page 7, line 206: authors state that SCO-spondin secreting cells varied in number along SCO length. What is the evidence that these cells secrete SCO-spondin? The staining shown in Figure 3L-O appears to demonstrate extracellular accumulation of sspo:GFP. What is the evidence that this staining originated from cells in proximity to it?The claim of SCO-secreting cells in Figure 2E-J is confusing. I assume you are using anatomy to infer the SCO is captured in these sections. This should be done in sspo-GFP animals (as in Figure 3) and/or dual anti-body labeling can be done to show SCO-secreting cells and cilia.

• We now show in Supplementary Figure 2 A-D a double staining for Sco-spondin-GFP and cilia (Ac-tub, Glu-Tub). Analyzing GFP staining along SCO length on successive sections, we identified the SCO producing cells on the diencephalic dorsal midline by their position under the posterior commissure (PC), which forms an Acetylated Tubulin positive arch, and counted the nuclei surrounded by cytoplasmic GFP from the most anterior region (24 cells wide, Sup Fig 2A-A’) to the most posterior region (4-8 cells wide, Sup Fig 2 C).`

• Furthermore, the close-ups presented on Fig 2A’ and 2B’ allow to detect the cytoplasmic Sspo-GFP staining around SCO nuclei, above the region presenting primary cilia pointing towards the diencephalic ventricle, both in controls and mutants at scoliosis onset (tail-up mutants), showing that the extracellular staining in B’ very likely originates from these cells. In these tail-up mutants, extracellular Sspo aggregates have not yet filled the whole diencephalic ventricle as in Fig 3 N and Q.

(9) Figure 5: Is the transcriptome data and proteomic data consistent for any transcripts and encoded protein products? Please highlight those consistent targets in both analyses.

• We would like to emphasize that the transcriptomic study was performed at scoliosis onset, at 5 weeks, while the proteomics analysis was performed at adult stage (3 months) so they cannot be directly compared.

Moreover, low abundance proteins (such as centrosomal proteins and transcription factors like Foxj1a) are not detected by label-free proteomics, without prior subcellular fractionation procedure (Lindemann et al, 2017 PMID: 28282288). The extraction protocol also does not allow to purify short neuropeptides such as Urp1-2.

Nevertheless, we found four targets in common, now highlighted in red in Fig 5, Panel E: Anxa2, complement proteins

C4 and C7a, and Stat3, all related to immune response, a GO term enriched in both studies as explained in the text (Lines 308-311, page 10).

The absence of many inflammation markers or immune response proteins at adult stage in scoliotic mutants most probably indicates a transient inflammatory episode at scoliosis onset, while astrogliosis, as detected by GFAP staining, increases with scoliosis severity. Along the same lines, the two-fold increase of Lcp1 cells within the tectum is present before axis curvature (in straight mutants) and disappears in scoliotic fish (Graph G in Sup Figure S5) as explained in the text, Lines 378-381, page 12,

(10) Supplementary Figure 1 F-H: What stage/age samples were used for SEM? It is only stated that they were 'adults'. It is also stated that cilia tufts in straight rpgrip1l-/- fish were morphologically normal but 'less dense'- this was not obvious from the figure. Can density be quantified? (otherwise, data does not support the statement). Similarly, can the statement that "cilia of mono-ciliated ependymal cells showed abnormal irregular structures compared to controls, with either bulged or thinner parts" be supported with measurements/quantification?

• The SEM study was performed on 3 months old fish, 3 controls and 5 mutants. We added this information in the figure legend. We could not quantify the number of ciliary tufts in the brain ventricle of the sole straight mutant that was analyzed. We therefore removed the statement that cilia were less dense in the straight mutant. Along the same lines, we mentioned that we could find mutant cilia of irregular shape as shown in Supplementary Figure S1, F”,G’’, H’’ and H’’’ (page 4, lines 124-129).

(11) Supplementary Figure 1D-E is never mentioned in the text. The Supplemental Figure legend also refers to a graph of cilia length that is not in the figure itself. As a result, many of the subsequent panel references are out of register.

• We now provide the correct version of the legend and refer to Sup Fig 1D-E in the text (page 3, lines 79-81) and its legend, page 53, lines 1616-1620.

(12) Supplementary Figure 2A-F: Of interest, in panels C and F, it looks as though sspo:GFP is accumulating on cilia within the ventricles of rpgrip1l mutants. Can this be explored? Is it possible that abnormal aggregation of SSPO on cilia is ultimately leading to cilia loss, as you report for multi-ciliated cells surrounding the subcommissural organ? This could be a very interesting finding and possible mechanism for cilia loss.

• Our observation of all brain sections led us to conclude that the majority of Sspo-GFP aggregates were floating within the brain ventricles of *rpgrip1l-*/- fish while a portion of aggregates were stuck on ventricle walls, in close contact with cilia as now shown on Supplementary figure S2 B’, outlined in legend page 54, lines 1634-1637. We agree that the contact between Sspo aggregates and cilia might have damaging consequences, either on cilia maintenance or on immune reaction induction and we now mention these possibilities in the discussion page16, lines 524-526. These research lines will be explored in the near future.

(13) Supplementary Figure 5A-F is not mentioned in the manuscript. Please clarify the role of Anxa2 in neuroinflammation. Is increased Anxa2 expression in rpgrip1l mutant zebrafish reduced after anti-inflammatory drug treatment? What is the expression level of anxa2 in cep290 mutant zebrafish?

• We have now added mention to Supplementary Figure 5A-F in the text page 10 lines 328-331.

• We unfortunately did not have enough histological material to test Anxa2 staining on NACET treated fish after performing GFAP and Lcp1 staining, neither for dilatation measurement or multiciliated cells quantification. We agree this would have helped to better define which defect might be an indirect consequence of an inflammatory environment.

• We tested the expression level of Anxa2 in *cep290-/-* fish. No labelling above control level was detected on *cep290-/-* brain sections that were positive for GFAP (N = 5). As GFAP staining in 3-4 weeks *cep290-/-* was not as intense and widespread as in adult *rpgrip1l-/-* (50% of GFAP + cells compared to 100% in the SCO for example), we concluded that Anxa2 expression may be upregulated after widespread or long-term astrogliosis/inflammation. Alternatively, Anxa2 overexpression could be specific to *rpgrip1l-/-* fish.

(14) A summary diagram at the end would be helpful for understanding the main findings.

We added a Graphical Abstract summarizing the main conclusions and hypotheses of this study. It is mentioned and explained in the Discussion section, p. 16 lines 504-508 and 516-529.

(15) The sspo-GFP zebrafish line should be listed in the STAR methods section:

The sspo-GFP line is now listed in the STAR methods, Scospondin-GFPut24, (Troutwine et al., 2020 PMID: 32386529), p.43, last line.